# *Plasmodium falciparum* parasites deploy RhopH2 into the host erythrocyte to obtain nutrients, grow and replicate

Natalie A Counihan[1], Scott A Chisholm[1], Hayley E Bullen[2], Anubhav Srivastava[3], Paul R Sanders[2], Thorey K Jonsdottir[2,4], Greta E Weiss[2], Sreejoyee Ghosh[1], Brendan S Crabb[2,4,5], Darren J Creek[3], Paul R Gilson[2,5], Tania F de Koning-Ward[1]*

[1]School of Medicine, Deakin University, Waurn Ponds, Australia; [2]Burnet Institute, Melbourne, Australia; [3]Drug Delivery, Disposition and Dynamics, Monash Institute of Pharmaceutical Sciences, Monash University, Parkville, Australia; [4]Department of Medicine, University of Melbourne, Parkville, Australia; [5]Monash University, Melbourne, Australia

**Abstract** *Plasmodium falciparum* parasites, the causative agents of malaria, modify their host erythrocyte to render them permeable to supplementary nutrient uptake from the plasma and for removal of toxic waste. Here we investigate the contribution of the rhoptry protein RhopH2, in the formation of new permeability pathways (NPPs) in *Plasmodium*-infected erythrocytes. We show RhopH2 interacts with RhopH1, RhopH3, the erythrocyte cytoskeleton and exported proteins involved in host cell remodeling. Knockdown of RhopH2 expression in cycle one leads to a depletion of essential vitamins and cofactors and decreased de novo synthesis of pyrimidines in cycle two. There is also a significant impact on parasite growth, replication and transition into cycle three. The uptake of solutes that use NPPs to enter erythrocytes is also reduced upon RhopH2 knockdown. These findings provide direct genetic support for the contribution of the RhopH complex in NPP activity and highlight the importance of NPPs to parasite survival.

*For correspondence: taniad@ deakin.edu.au

**Competing interests:** The authors declare that no competing interests exist.

## Introduction

Malaria is caused by infection of the blood with Apicomplexan parasites of the genus *Plasmodium*. Critical for the proliferation and survival of *Plasmodium* in the blood is their ability to quickly penetrate host erythrocytes and acquire nutrients required for rapid growth. To facilitate this, the invasive merozoite forms of *Plasmodium* spp. sequentially secrete proteins from their apical organelles, the micronemes, rhoptries and dense granules. Proteins localizing to the micronemes and rhoptry neck are implicated in the irreversible attachment of the parasite to the host cell and are critical for invasion (reviewed in [*Harvey et al., 2012*; *Weiss et al., 2016*]). Dense granule proteins are secreted once *Plasmodium* parasites have invaded their host cell (*Riglar et al., 2011*), contributing to remodeling of the host cell (*de Koning-Ward et al., 2016*). However, the role of proteins that localize to the rhoptry bulb is less clear and although they have been implicated in roles ranging from rhoptry biogenesis, erythrocyte invasion, formation of the parasitophorous vacuole (PV) in which the parasite is encased, as well as modification of the host cell (*Kats et al., 2006*; *Counihan et al., 2013*), functional data supporting these roles is very limited.

RhopH2 is one of ~15 known proteins that localize to the rhoptry bulb in *Plasmodium* merozoites (*Counihan et al., 2013*; *Ling et al., 2003*). It is found in a high molecular weight complex with RhopH1 and RhopH3 (*Cooper et al., 1988*) that is discharged from merozoites, associating with the erythrocyte surface upon merozoite contact (*Sam-Yellowe et al., 1988*; *Sam-Yellowe and Perkins,*

**eLife digest** Malaria is a life-threatening disease that affects millions of people around the world. The parasites that cause malaria have a complex life cycle that involves infecting both mosquitoes and mammals, including humans. In humans, the parasites spend part of their life cycle inside red blood cells, which causes the symptoms of the disease. In order to thrive, malaria parasites need to make the red blood cell more permeable so that they can absorb nutrients from the blood stream and get rid of toxic waste products they generate.

Previous research has shown that the parasites can produce a protein that makes red blood cells more permeable to a range of nutrients. Understanding how the parasites can do this, and if they could change the permeability of their host red blood cell to prevent anti-malaria drugs from entering may help researchers to identify new approaches to starve the parasite.

Counihan et al. investigated whether the parasites also use other proteins to modify red blood cells and demonstrated that a protein called RhopH2 can make the blood cells more permeable. The experiments used a genetically modified version of the parasite that lacked RhopH2. Counihan et al. show that essential nutrients and vitamins were depleted in these parasites and that the parasites were much slower to grow and reproduce.

The next important step would be to identify all proteins that are involved in making red blood cells more permeable and how they achieve this, and use this knowledge to help generate anti-malarial drugs.

*1991*). The localization of RhopH proteins in the newly-infected erythrocyte is less clear as multiple localizations, including the PV membrane (PVM), Maurer's clefts and the cytosolic face of the erythrocyte membrane have been described for its constituents using different experimental approaches (*Perkins and Ziefer, 1994*; *Ndengele et al., 1995*; *Sam-Yellowe et al., 2001*; *Hiller et al., 2003*; *Vincensini et al., 2005*, *2008*). RhopH2 and RhopH3 are each encoded by a single gene. In contrast, RhopH1 in *P. falciparum,* the most pathogenic of the species infecting humans, is encoded by a multi-gene family comprising five variant genes termed *clag 2, 3.1, 3.2, 8* and *9* (with *clag3.1* and *3.2* mutually exclusively transcribed) (*Gupta et al., 2015*; *Kaneko et al., 2001*, *2005*; *Ling et al., 2004*).

Of all the RhopH proteins, putative functions have only been assigned for RhopH1/Clag3 and Clag9 (*Gupta et al., 2015*), although there is conflicting evidence for the involvement of Clag9 in cytoadherence (*Trenholme et al., 2000*; *Goel et al., 2010*; *Nacer et al., 2011*). Via a high throughput drug-screening approach Clag3 has been linked to plasmodial surface anion channel (PSAC) activity (*Nguitragool et al., 2011*). PSAC is a type of new permeability pathway (NPP) induced in the erythrocyte membrane by *Plasmodium* spp. that increases the cell's porosity to organic and inorganic solutes. *P. falciparum* Clag3 null-mutants exhibit delayed in vitro growth, although NPP activity has not been investigated (*Comeaux et al., 2011*). Intriguingly, Clag3 exhibits no homology to known ion channel proteins and lacks conventional membrane spanning regions to form a pore through the erythrocyte membrane, although it exists as both an integral and peripheral membrane protein in the infected erythrocyte (*Nguitragool et al., 2011*; *Zainabadi, 2016*). Thus whether Clag3 forms ion channels directly and exclusively or if other parasite proteins or host cell membrane components contribute to a functional NPP is unknown. Alternatively, Clag3 may participate indirectly, for example, by regulating NPP activity.

Both the *rhopH2* gene and *rhopH3* gene are refractory to deletion (*Cowman et al., 2000*; *Janse et al., 2011*). As RhopH1 is encoded by a multi-gene family, it is difficult to establish without genetically disrupting all but one *clag* variant within a parasite, whether the *clag* genes serve complementary functions or play distinct roles, including in NPP activity. To address these questions, we characterized RhopH2 in *P. falciparum* and conditionally depleted its expression in *P. falciparum* and the rodent malaria parasite *P. berghei* to investigate its contribution to erythrocyte invasion, parasite growth and erythrocyte permeability. Depletion of RhopH2 in cycle one did not affect transition into cycle two, suggesting RhopH2 plays no direct role in invasion. However, NPP activity was greatly reduced and parasite growth slowed as parasites progressed into trophozoite stage in cycle two, possibly due to nutrient depravation and/or accumulation of waste products. Transition into cycle

three was curtailed by interesting phenomena including reduced schizont rupture and merozoite malformation that may be linked to reduced de novo pyrimidine synthesis. Taken together, RhopH2 appears to be important for NPP activity and for the exchange of nutrients and wastes with the blood plasma to facilitate parasite growth and proliferation.

## Results

### Modification of the *rhoph2* locus in *P. falciparum*

Conditional gene knockdown approaches were utilized herein to gain insight into the functional role of RhopH2 in *Plasmodium* parasites. This involved transfecting pRhopH2-HAglmS into *P. falciparum* that when correctly integrated into the *rhopH2* locus, would lead to incorporation of a triple hemagglutinin (HA) and single strep II tag at the C-terminus of RhopH2 and the glucosamine (GlcN)-inducible *glmS* ribozyme (*Prommana et al., 2013*) within its 3' untranslated region (UTR) (*Figure 1a*). Diagnostic PCR of transfectants resistant to WR99210 selection after three rounds of drug cycling confirmed that transgenic parasites, termed PfRhopH2-HAglmS, harbored the expected integration event (*Figure 1b*). This was further validated by western blotting of parasite lysates from clonal PfRhopH2-HAglmS parasites using an anti-HA antibody; RhopH2 typically runs at 140 kDa by SDS-PAGE (*Cooper et al., 1988*; *Ling et al., 2003*) and the observed 150 kDa band of RhopH2-HA is consistent with its anticipated size (*Figure 1c*). Immunofluorescence analysis (IFA) confirmed RhopH2-HA localized to the rhoptry and co-localized with other rhoptry bulb proteins, RhopH1, RhopH3 and RAMA but not with the rhoptry neck protein, RON4, the micronemal marker, AMA-1 or the plasma membrane protein MSP1 (*Figure 1d*). Comparison of the wildtype 3D7 and RhopH2-HAglmS parasite lines revealed that the addition of the epitope tags and ribozyme sequence did not impact on RhopH2-HAglmS to grow normally (*Figure 1—figure supplement 1*).

### RhopH2 migrates from the PVM to the erythrocyte periphery after invasion

As RhopH2 has been described to reside at several different locations post-invasion, we took advantage of our RhopH2-HA line to characterize the expression and localization of RhopH2 at different times post-invasion using anti-HA antibodies. Western blot analysis revealed weak expression of RhopH2 during the ring and trophozoite stages, with a peak of expression at schizont stage (*Figure 2a*), in keeping with when RhopH2 is maximally transcribed (*Ling et al., 2003*). IFA confirmed RhopH2 synthesized during schizogony is carried in during invasion and localizes to the interface between the parasite and host cell (*Figure 2b*). Weak labeling could also be observed at the erythrocyte membrane. As the parasite matured, RhopH2 could be detected in the erythrocyte cytoplasm, often exhibiting distinct punctate labeling, and the intensity of labeling at the erythrocyte membrane became more pronounced. RhopH2 did not co-localize with SBP1, a Maurer's cleft resident protein, indicating RhopH2 is not trafficked to the erythrocyte membrane via these membranous structures as previously suggested (*Sam-Yellowe et al., 2001*; *Vincensini et al., 2005*) (*Figure 2c*).

Although RhopH2 has been shown to be present in detergent resistant membranes at schizont stages (*Sanders et al., 2007*; *Hiller et al., 2003*), localizes to the erythrocyte cytosolic face of the PVM (*Hiller et al., 2003*) and is present at the erythrocyte membrane, it is unclear how RhopH2 associates with these membranes. The hydrophobic region at I739-H759 is not universally predicted as a conventional transmembrane domain bioinformatically (eg. TMHMM, SOSUI, TMPred). Therefore, we examined the solubility profile of RhopH2 at both schizont and ring-stages. We found that in contrast to EXP2, which is a component of the *Plasmodium* translocon of exported proteins (PTEX) that resides at the PVM and requires Triton X-100 to be extracted from the membranes (*de Koning-Ward et al., 2009*), the majority of RhopH2 could already be extracted with carbonate when sequential solubility assays were conducted (*Figure 2d*, top panel). This indicates that RhopH2 is peripherally associated with membranes and is not an integral membrane protein. However, when erythrocytes infected with ring-stage parasites were saponin-lysed, pelleted by centrifugation and resuspended directly (rather than sequentially) in various detergents/buffers, RhopH2 could be extracted with carbonate and mostly with urea (which also extracts peripheral membrane proteins), whereas it remained in the Triton X-100 pellet fraction (*Figure 2d*, bottom panel). Combined, this

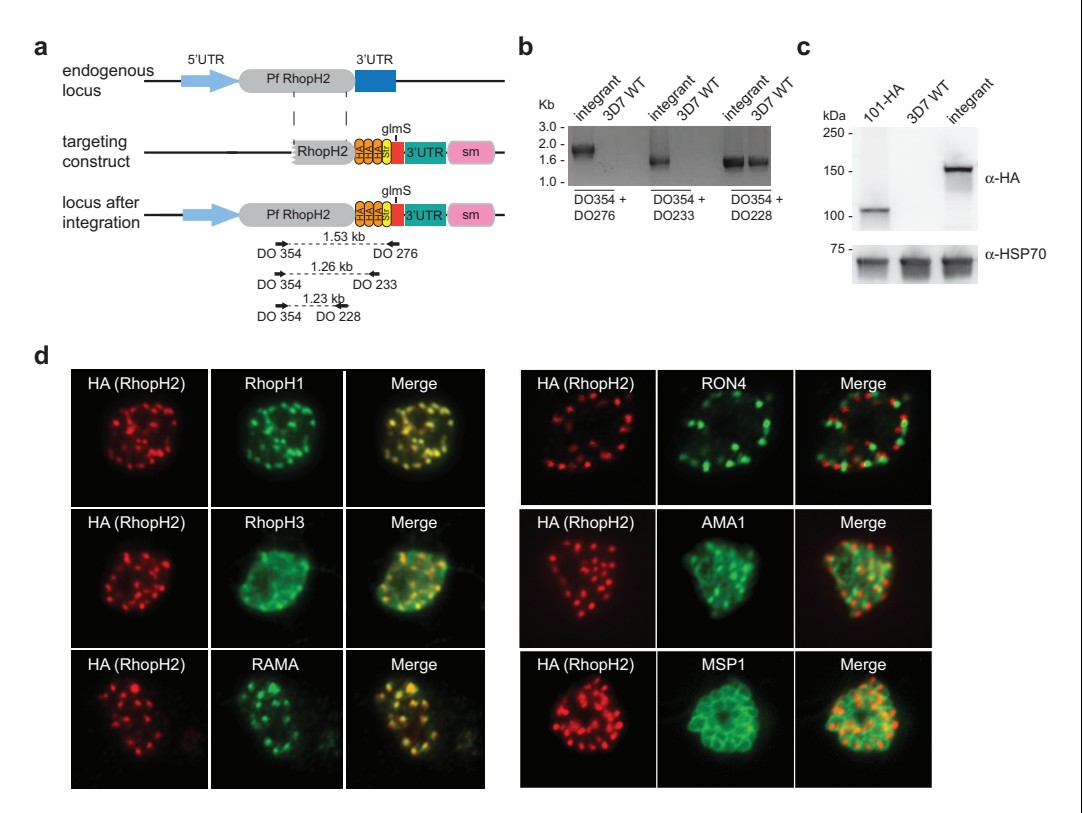

**Figure 1.** Generation of transgenic parasites in which RhopH2 is epitope-tagged. (**a**) The *P. falciparum* RhopH2 targeting construct was designed to integrate into the endogenous locus by a single crossover recombination event. The predicted structure of the endogenous locus before and after integration is shown. Haemagglutinin (HA) and strep II (Str) epitope tags, selectable marker (sm), *glmS* ribozyme and untranslated regions (UTR) are shown. Arrows indicate oligonucleotides used in diagnostic PCR analysis and indicative product size. (**b**) Diagnostic PCR showing the *PfRhoph2* gene contains the integrated sequence. Oligonucleotide pairs shown in (**a**) were used on genomic DNA prepared from drug-resistant parasites after transfection with the targeting construct (integrant) or 3D7 (WT). DO354 and DO228 oligonucleotides, which recognize the *rhoph2* locus, serve as a positive control for the PCR. (**c**) Western blot analysis showing the integrant line expresses the HA epitope tags. The predicted molecular mass of epitope-tagged RhopH2 is 164 kDa. PfHSP101-HA (101-HA) serves as a positive control. (**d**) Immunofluorescence analysis (IFA) on schizonts fixed with acetone/methanol and labelled with anti-HA antibody to detect RhopH2 and other antibodies, as indicated.

The following figure supplement is available for figure 1:

**Figure supplement 1.** Comparison of growth between *P. falciparum* wildtype (3D7) and RhopH2-HAglmS parasite lines.

data indicates that while RhopH2 predominantly has a peripheral association with the membranes at its respective locations, RhopH2 may be interacting with erythrocyte cytoskeletal proteins or is present in lipid rafts during the ring-stages, leading to its insolubility in Triton X-100 when resuspended in this buffer directly.

## RhopH2 interacts with exported proteins and components of the erythrocyte cytoskeleton

To gain insight into proteins that interact with RhopH2 post-invasion, we next investigated the interactome of RhopH2 in ring and trophozoite stages by immunoprecipitating RhopH2 from PfRhopH2-HAglmS lysates using anti-HA antibodies and identifying proteins that had been affinity purified by mass-spectrometry (*Figure 3a*). Bead-only and irrelevant protein controls (*Elsworth et al., 2016*) were used to identify non-specific interactions including ribosomal, nuclear and cytosolic proteins, which were subtracted to attain a list of likely specific interactions.

In both parasite stages, RhopH2 was pulled down as well as other members of the RhopH complex, relatively few PV proteins such the PTEX complex, many exported PEXEL proteins, especially in

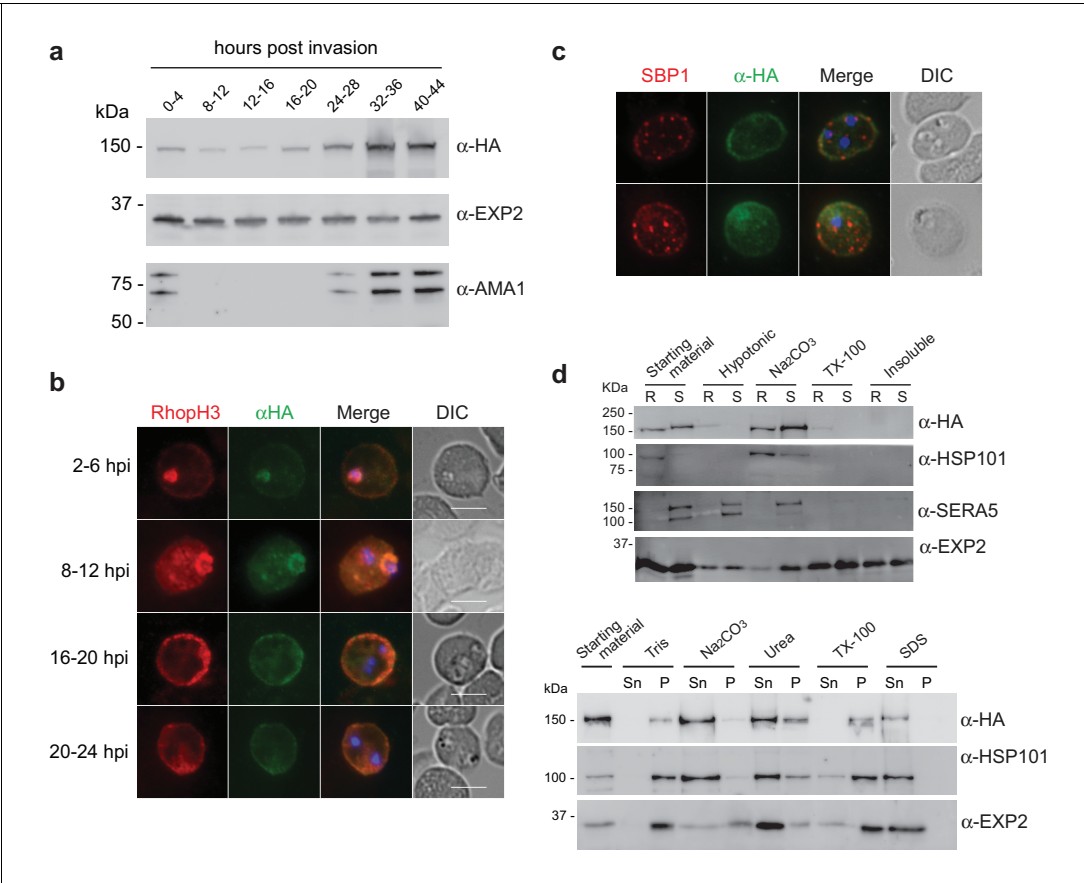

**Figure 2.** Expression, localisation and solubility profile of *P. falciparum* RhopH2. (**a**) Western blot analysis of RhopH2-HA expression across the erythrocytic cycle. Immunoblots were probed with the antibodies as indicated. (**b**) Immunofluorescence analysis (IFA) on erythrocytes infected with PfRhopH2-HAglmS and fixed with acetone/methanol. RhopH2 is labeled with the anti-HA antibody. The bars represent 5 µm. (**c**) IFA on erythrocytes infected with PfRhopH2-HAglmS, fixed with acetone/methanol and probed with anti-HA (for RhopH2) and antibodies to the Maurer's cleft protein SBP1 show that RhopH2 and SBP1 do not co-localise. (**d**) Solubility of RhopH2-HAglmS. Upper panel: Infected erythrocytes were synchronized and saponin-lysed when parasites reached ring (R) or schizont (S) stage and the pelleted material was sequentially dissolved in the buffers as indicated in the order of left to right (upper panel). Supernatant fractions were analysed by western blotting with the indicated antibodies. Insoluble material represents protein remaining in the pellet fraction after 1% Triton X-100 treatment. Lower panel: Alternatively, infected erythrocytes were saponin-lysed when parasites were at ring stages, split into equal portions and pelleted before dissolving in one of the indicated buffers. Both supernatant (Sn) and pellet (P) fractions were analysed by western blotting.

trophozoites, and a large number of erythrocyte cytoskeletal proteins, particularly in the ring stages (*Figure 3b*). That RhopH2 was interacting with the other members of the RhopH complex (*Figure 3c*) is consistent with an earlier report demonstrating the RhopH complex persists intact for at least 18 hr post-invasion (*Lustigman et al., 1988*). There were more peptides recovered for RhopH1 (particularly Clags3.1, 3.2 and 9) and RhopH3 than there were for RhopH2 at the ring and trophozoite stages (*Figure 3c*). Given that predicted molecular weights for the Clags (160–171 kDa) and RhopH2 (163 kDa) are similar but that of RhopH3 is somewhat smaller (104 kDa), this indicates that each component of the RhopH complex may not be in a 1:1:1 stoichiometry. Blue-Native PAGE gel analysis revealed that RhopH2 is present in a ~670 kDa complex that has a molecular mass larger than the predicted ~425 kDa (*Figure 3d*). Apart from not being in an equimolar ratio, other non-RhopH proteins may also be present in the ~670 kDa complex. We also observed a smaller ~410 kDa complex when using the zwitterionic detergent 3-(tetradecanoylamidopropyl dimethylammonio) propane 1-sulfonate (ASB-14), which could contain a subset of the RhopH proteins and/or other proteins (*Figure 3d*).

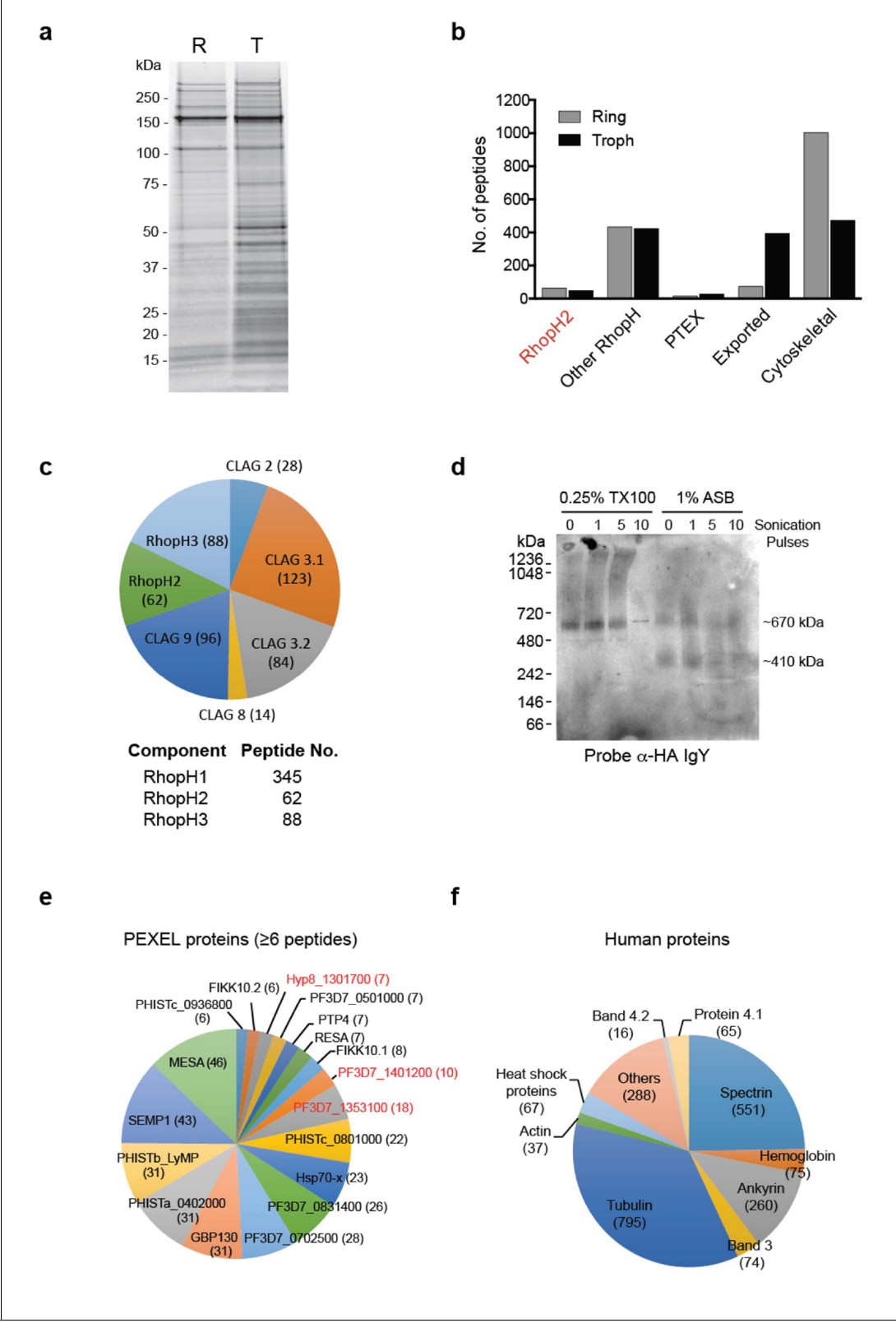

**Figure 3.** The RhopH2 interactome. (**a**) Coomassie-stained SDS-PAGE gel of elution fractions from immune-precipitations performed with HA antibodies on lysates made from erythrocytes infected with RhopH2-HAglmS parasites at ring (R) or trophozoite stage (T). (**b**) Bar graph showing the total number of peptides of particular subclasses of proteins that were affinity purified with PfRhopH2-HA. (**c**) Pie charts showing the number of peptides from the respective RhopH proteins that affinity purified with RhopH2-HA. The numbers of peptides identified are indicated in brackets. Note

*Figure 3 continued on next page*

Figure 3 continued

RhopH1 includes all CLAG peptides. (d) Western blot of blue-native PAGE performed on erythrocytes infected with trophozoite stage RhopH2-HAglmS parasites that had been solubilized in either 0.25% Triton X-100 or 1% ASB detergent reveal RhopH2 is present in ~670 and ~410 kDa species. (e) Pie chart showing the numbers of the most abundant peptides from PEXEL proteins that affinity purified with PfRhopH2-HA from trophozoite stage parasites. (f) Pie chart showing the numbers of the most abundant peptides from host erythrocyte proteins that affinity purified with PfRhopH2-HA in ring stage parasites.

Almost as abundant as RhopH peptides identified from the trophozoite-stage immunoprecipitations were exported proteins (*Figure 3b and e*). The most predominant peptides from known exported proteins included those of mature parasite-infected erythrocyte surface antigen (MESA; a protein that interacts with host protein 4.1) (*Waller et al., 2003*), small exported membrane protein 1 (SEMP1; a non-essential protein that localizes to the Maurer's clefts and is partially translocated to the erythrocyte membrane) (*Dietz et al., 2014*), glycophorin-binding protein 130 (GBP130; an exported soluble protein) (*Maier et al., 2008*), a variety of *Plasmodium* helical interspersed subtelomeric proteins (PHIST; some of which have been shown to interact with PfEMP1) (*Proellocks et al., 2014*; *Oberli et al., 2014*, *2016*) and HSP70-x (localizes to J-dots) (*Külzer et al., 2012*). Peptides from exported proteins were more abundant in the pull-down performed on lysates from trophozoites compared to ring-stages, in keeping with protein export peaking during the trophozoite stage. The exception was ring erythrocyte surface antigen (RESA) in which more peptides were observed in the ring-stage pull-down. RESA is one of the first proteins exported into the erythrocyte that ultimately localizes to the ankyrin-band 3 complex at the erythrocyte cytoskeleton. A large number of peptides to erythrocyte cytoskeletal proteins were also identified, including tubulin, spectrin, ankyrin, protein 4.1, band 3 and actin (*Figure 3f*). Whether RhopH2 is indirectly interacting with these cytoskeletal components via exported proteins, or specifically interacting with all or a subset of these cytoskeletal proteins is unknown, especially since many cytoskeletal elements are bound together in the cell. Taken together, these results indicate that after invasion, the RhopH complex traffics from the PVM to the erythrocyte membrane and either en route or when it reaches its final destination, RhopH2 interacts with a number of exported parasite proteins that also bind to components of the host cytoskeleton.

## Knockdown of RhopH2 reduces parasite growth and proliferation

As the epitope-tagged RhopH2 line harbors a *glmS* riboswitch sequence, the ability to regulate RhopH2 expression in parasites via the addition of GlcN was investigated to gain functional insight into this protein. Erythrocytes infected with synchronized ring-stage parasites were treated for up to two cycles with 2.5 mM GlcN and assessed for protein knockdown via western blot (*Figure 4a*) and parasite growth via Giemsa-stained smears relative to parasites grown in the absence of GlcN (*Figure 4b–c*). RhopH2 is normally transcribed around the onset of schizogony (*Ling et al., 2003*; *Bozdech et al., 2003*; *Le Roch et al., 2004*) and the addition of 2.5 mM GlcN resulted in knockdown of RhopH2 expression in schizonts by 84% within the first cycle. By late in the second cycle, RhopH2 protein levels were reduced by 92% in those parasites that made it to schizont stage (*Figure 4a*). RhopH2-HAglmS parasites (+GlcN) appeared morphologically normal by the end of the first cycle (*Figure 4b*). In separate experiments whereby RhopH2-HAglmS parasites expressing GFP at the end of cycle one were incubated with donor erythrocytes, the conversion of schizonts to ring stage parasites and therefore invasion efficiency was not specifically affected by the knockdown of RhopH2 (*Figure 4c*). In contrast, a striking growth defect in RhopH2-HAglmS (+GlcN) parasites was observed in the second cell cycle around the ring to trophozoite transition stage, with late-ring stage parasites appearing irregular in shape and trophozoites exhibiting an abnormal stunted phenotype rather than progressing to mature trophozoites (*Figure 4b*). In addition, RhopH2-HAglmS (+GlcN) parasites that transitioned to schizonts at the end of the second cycle harboured significantly lower numbers of merozoites per schizont (mean of 19 merozoites cf 12 merozoites for –GlcN and +GlcN cultures, respectively p<0.0001) (*Figure 4d*). Moreover, the time required to complete the second cycle and commence the third cycle was delayed (~92 hr cf ~108 hr for -GlcN and +GlcN cultures, respectively) (*Figure 4b,e*). This all translated to ~4 fold reduction in the number of ring-stage parasites observed at the beginning of the third cycle when compared to parasites not exposed to GlcN (*Figure 4e*).

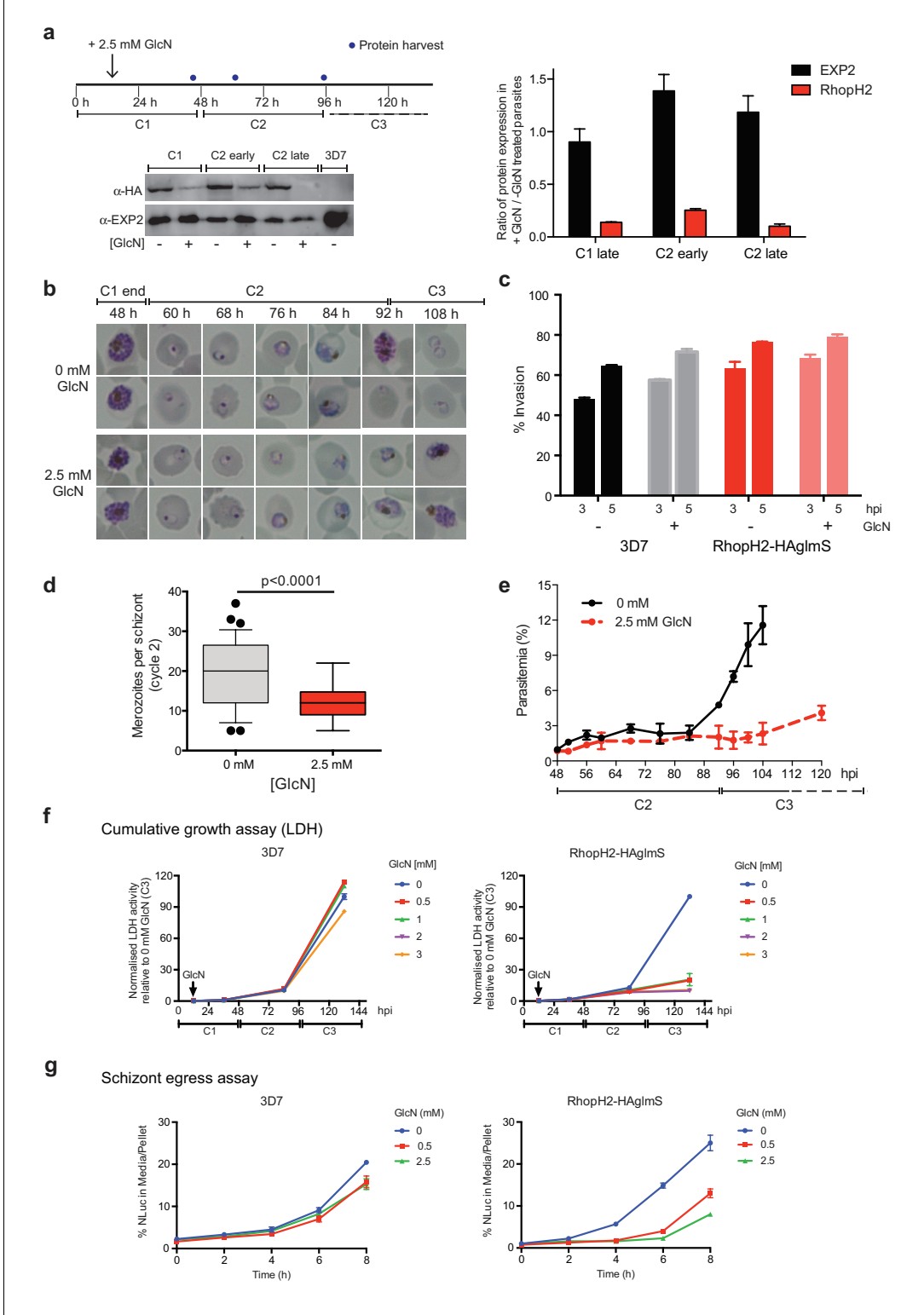

**Figure 4.** Reduction in PfRhopH2 expression leads to altered growth phenotypes in vitro. (a) Effect of glucosamine on PfRhopH2 protein expression. Upper panel: overview of experiment. Synchronised cultures of PfRhopH2-glmS were treated with glucosamine (GlcN) at the indicated time and material harvested, as indicated. Lower panels: infected erythrocytes were harvested by saponin lysis and subject to SDS-PAGE and western blotting. PfRhopH2-HA was detected using an anti-HA antibody and EXP2 (used as a loading control) detected with a specific polyclonal EXP2 antibody. Right panel: Densitometry performed on bands observed in western blot using ImageJ was performed to calculate the ratio of EXP2 or RhopH2 protein levels in parasite lines grown in the presence (+) or absence (-) of GlcN (n = 3 independent experiments). Shown is the mean ± SEM (n = 3). (b) Representative

*Figure 4 continued on next page*

*Figure 4 continued*

Giemsa-stained smears parasites depleted of RhopH2 progress to schizont stage in cycle one but parasite growth is slowed around the trophozoite stage (n = 3 independent experiments). (c) Analysis of the number of schizonts in cultures of wildtype (3D7) and RhopH2-HAglmS parasites grown in the absence (−) or presence (+) of 2.5 mM GlcN that invaded donor erythrocytes within 3 or 5 hr post-incubation (hpi), as measured by FACS (n = 3). Shown is the mean ± SEM. (d) Box plot indicating the number of merozoites formed per schizont in cultures of RhopH2-HAglmS grown in 0 mM (35 schizonts examined) or 2.5 mM (51 schizonts examined) GlcN. The central bar in the box plot denotes the median whilst the whiskers delineate the 10th and 90th percentiles. p<0.0001 by unpaired t-test. (e) Parasitemias of cultured PfRhopH2-HAglmS parasites grown in 0 mM or 2.5 mM GlcN, determined by counting a minimum of 1000 erythrocytes. Depletion of PfRhopH2 expression increases the length of the cell cycle and has a marked effect on the numbers of parasites progressing to cycle 3. Shown is the mean ± SEM (n = 3). (f) Growth of 3D7 and PfRhopH2-HAglmS parasites when cultured in various concentrations of GlcN, as measured by lactate dehydrogenase assay (LDH). The LDH activities of 3D7 and RhopH2-HAglmS cultured in the absence of GlcN at cycle three were normalized to 100%, and activity of all lines (± GlcN) across the three cycles was measured relative to this. Shown is the mean ± SD (n = 3). An unpaired t-test revealed RhopH2-HAglmS parasites grew significantly slower than 3D7 in all concentrations of GlcN by 36 hpi (p<0.01) (g) Measurement of nanoluciferase (Nluc) released into the culture media and in pelleted erythrocytes infected with 3D7 or RhopH2-HAglmS parasites expressing Hyp1-Nluc. Measurements commenced around the time 3D7 parasites were starting to egress and invade new erythrocytes. The data represents the mean ± SD of one biological replicate completed in triplicate, with results expressed as percentage Nluc activity in the media relative to the pellet fraction.

In separate experiments, parasite lactate dehydrogenase (pLDH) activity was also measured on trophozoite stages of RhopH2-HAglmS- or 3D7-parasitized erythrocytes grown in the presence of increasing concentrations of GlcN as a surrogate for parasite proliferation (*Figure 4f*). The pLDH activities of 3D7 (-GlcN) and RhopH2-HAglmS (-GlcN) at cycle three were normalized to 100%, with activity of all parasite lines (± GlcN) across the three cycles measured relative to this. While 3D7 parasite growth only begun to be affected by the addition of 2 mM GlcN by cycle 3, in strong contrast, growth of RhopH2-HAglmS parasites was majorly reduced at all GlcN concentrations and also relative to 3D7 cultured in the same GlcN concentrations. The results also concur with the experiments above in that the effect of GlcN on pLDH activity could already be seen at cycle two and was drastically amplified when parasites transitioned into cycle three (*Figure 4f*).

To further validate the effects of knocking down RhopH2 upon parasite maturation and transition into cycle three, erythrocytes infected with RhopH2-HAglmS and the 3D7 parental line at trophozoite stage were transfected with an exported nanoluciferase fusion protein (Hyp1-Nluc). This enabled schizont rupture and merozoite egress at the end cycle two to be monitored via measuring the amount of nanoluciferase released into the culture media compared to the cell pellet. Infected erythrocytes were supplemented with GlcN in cycle one when the parasites were at trophozoite stage and when the 3D7 line had grown to late-schizont stage in cycle two and new ring-stage parasites were beginning to be visible in Giemsa stain (indicative of the start of merozoite egress and commencement of transition into cycle 3), the media and cell pellets were harvested and every two hours thereafter for eight hours. The percentage ratio of nanoluciferase activity of media/pellet was then determined. This revealed that egress of RhopH2-HAglmS (+GlcN) line was markedly delayed compared to 3D7 (+GlcN) (p<0.001 for 2.5 mM GlcN) (*Figure 4g*). These results validate that the slower growth of RhopH2-HAglmS (+GlcN) observed in cycle 2 is due to the specific effect of depletion of RhopH2 expression.

## Knockdown of RhopH2 produces a profound defect in the invasion capacity of cycle two merozoites

Since the growth experiments revealed a defect in parasite transition from cycle two to cycle three, video microscopy of live schizont-stage parasites at the end of the second cycle on GlcN was performed to visualize whether knockdown of RhopH2 was impacting on erythrocyte egress and invasion. No obvious differences were observed in general schizont morphology or in the ability of the erythrocyte to burst, indicating egress *per se* was not actually affected. Rather, instead of the merozoites dispersing rapidly after egress, GlcN treatment caused the merozoites to remain clumped together (*Figure 5a*, *Videos 1–2*), a phenotype not observed in 3D7 (+GlcN) parasites (not shown). Occasionally remnants of membranes could be observed around the merozoites, but even when these broke down, the merozoites remained clumped (*Figure 5a*, see 3 mM 5s versus 29.1s). Nine and eleven schizont ruptures were observed ± GlcN treatment respectively, and as a consequence of merozoite clumping following GlcN treatment, less than two merozoites per rupture were released

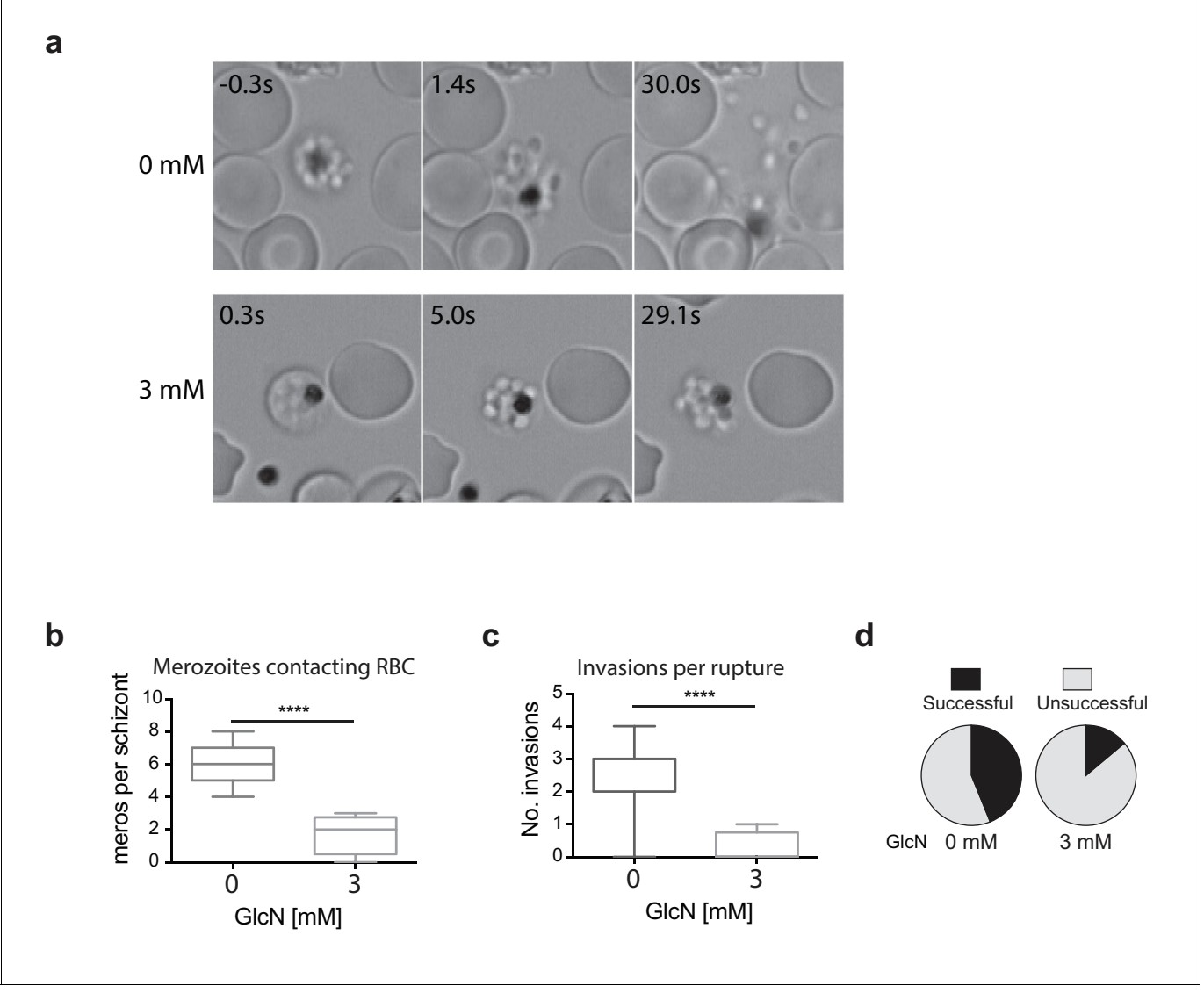

**Figure 5.** Merozoites depleted of PfRhopH2 show defect in parasite invasion the following cycle. (a) Panel of images from videos of PfRhopH2-HAglmS schizonts observed rupturing and releasing merozoites at the end of cycle 2, post-addition of 0 or 3 mM GlcN. The number of seconds post-rupture is indicated. (b) The number of merozoites contacting nearby erythrocytes per schizont rupture following GlcN treatment is shown. (c) The number of erythrocyte invasions per schizont rupture is shown. (d) The proportion of merozoite-erythrocyte contacts that successfully result in invasion are indicated. For (b) and (c), the central bar denotes median, the box denotes 25–75th percentile and the whiskers the data range. ****p<0.0001 by unpaired t-test.

and able to contact new erythrocytes compared with six merozoites without GlcN (*Figure 5b*). The net effect was fewer average invasions per rupture, with only 0.25 after GlcN treatment compared to 2.6 without treatment (*Figure 5c*). Whilst 28 out of the 63 merozoites that made erythrocyte contact without GlcN went on to invade erythrocytes (*Figure 5d*), only two out of the 14 merozoites treated with GlcN invaded erythrocytes, indicating a success rate of 0.44 and 0.14 invasions per contact, respectively (*Figure 5d*). From these results, it was inferred that the lower fold-increase in parasitemia from cycle two to the next after RhopH2 knockdown stemmed from a combined effect of reduction in the number of parasites reaching schizogony in cycle two and a reduced invasion rate. The latter most likely stems from an indirect effect of RhopH2 knockdown that results in a clumping of merozoites incapable of breaking free to invade a new host cell and a reduced competency of merozoites forming at the end of cycle two to successfully invade an erythrocyte.

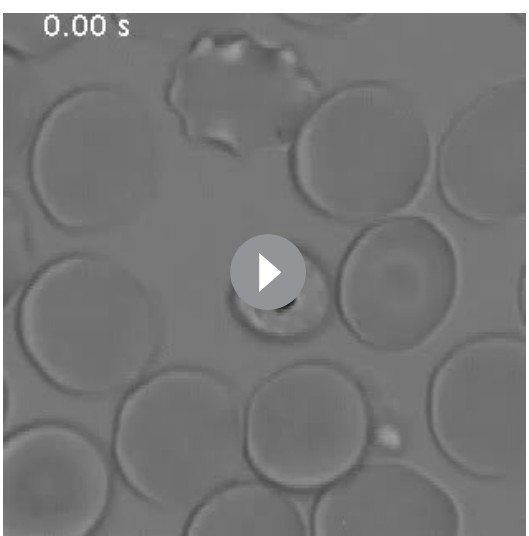

**Video 1.** *Plasmodium falciparum* RhopH2-HAglmS schizont rupturing and releasing merozoites which invade nearby human erythrocytes. Successful invasions are indicated with white arrows. Time in seconds from egress is indicated.

**Video 2.** A rupturing *Plasmodium falciparum* RhopH2-HAglmS schizont that had been treated with 3 mM glucosamine for 2 cell cycles to knockdown RhopH2-HA expression. At 0 s the erythrocyte membrane surrounding the schizont begins to break down but the merozoites do not disperse until about 68 s later. None of the merozoites appeared to invade neighbouring erythrocytes.

## Modification of the *rhoph2* locus in *P. berghei* affects parasite growth in vivo

To unequivocally show that the growth defects in *P. falciparum* were a consequence of RhopH2 knockdown, conditional regulation of RhopH2 in *P. berghei* was also performed. This also provided insight into the consequences of depleting RhopH2 expression on parasite growth in vivo. In this case, the *P. berghei rhoph2* locus was modified to insert an anhydrotetracycline (ATc)-regulated transactivator element (TRAD) downstream of the endogenous *rhoph2* promoter and a minimal promoter with TRAD binding sites upstream of the *rhoph2* coding sequence. *P. berghei* ANKA parasites transfected with linearized pTRAD4-RhopH2ss and surviving pyrimethamine drug pressure were analyzed by diagnostic PCR and Southern blot, confirming that the targeting construct had integrated correctly into the *rhoph2* locus and the line was clonal (*Figure 6—figure supplement 1a–c*). Transcription of *rhoph2* in this line, termed PbRhopH2-iKD, was highly responsive to ATc, with ~11-fold reduction of *rhoph2* mRNA in schizont stages as determined by qRT-PCR and RT-PCR (*Figure 6—figure supplement 1d*).

The growth of the PbRhopH2-iKD line was specifically sensitive to ATc treatment. PbRhopH2-iKD parasites grew poorly in mice that had been pre-exposed to ATc 24 hr prior to infection (*Figure 6a*). Conversely, growth of parental *P. berghei* parasites was unaffected by the presence of ATc (*Figure 6a*) as has been shown previously (*Pino et al., 2012*; *Elsworth et al., 2014*). The slower growth of PbRhopH2-iKD exposed to sucrose compared to parental *P. berghei* parasites exposed to ATc is most likely due to the transactivator not being able to induce transcription of *rhopH2* to the same level as the native promoter.

Since the more mature stages of *P. berghei* sequester in vivo, erythrocytes infected with PbRhopH2-iKD parasites ($1 \times 10^7$) were inoculated into ATc-pretreated mice and harvested the following cycle when the parasites were at ring-stage. They were then cultured ex vivo in the presence of ATc to examine the development of parasites across the entire cell cycle. Parasites in which RhopH2 had been depleted, exhibited delayed progression to trophozoite stage and the schizont stages displayed aberrant morphology, often appearing vacuolated and containing fewer merozoites (*Figure 6b–c*). A synchronous in vitro invasion and growth assay using merozoites that had been generated from cultured schizonts confirmed these findings, showing that parasites depleted of

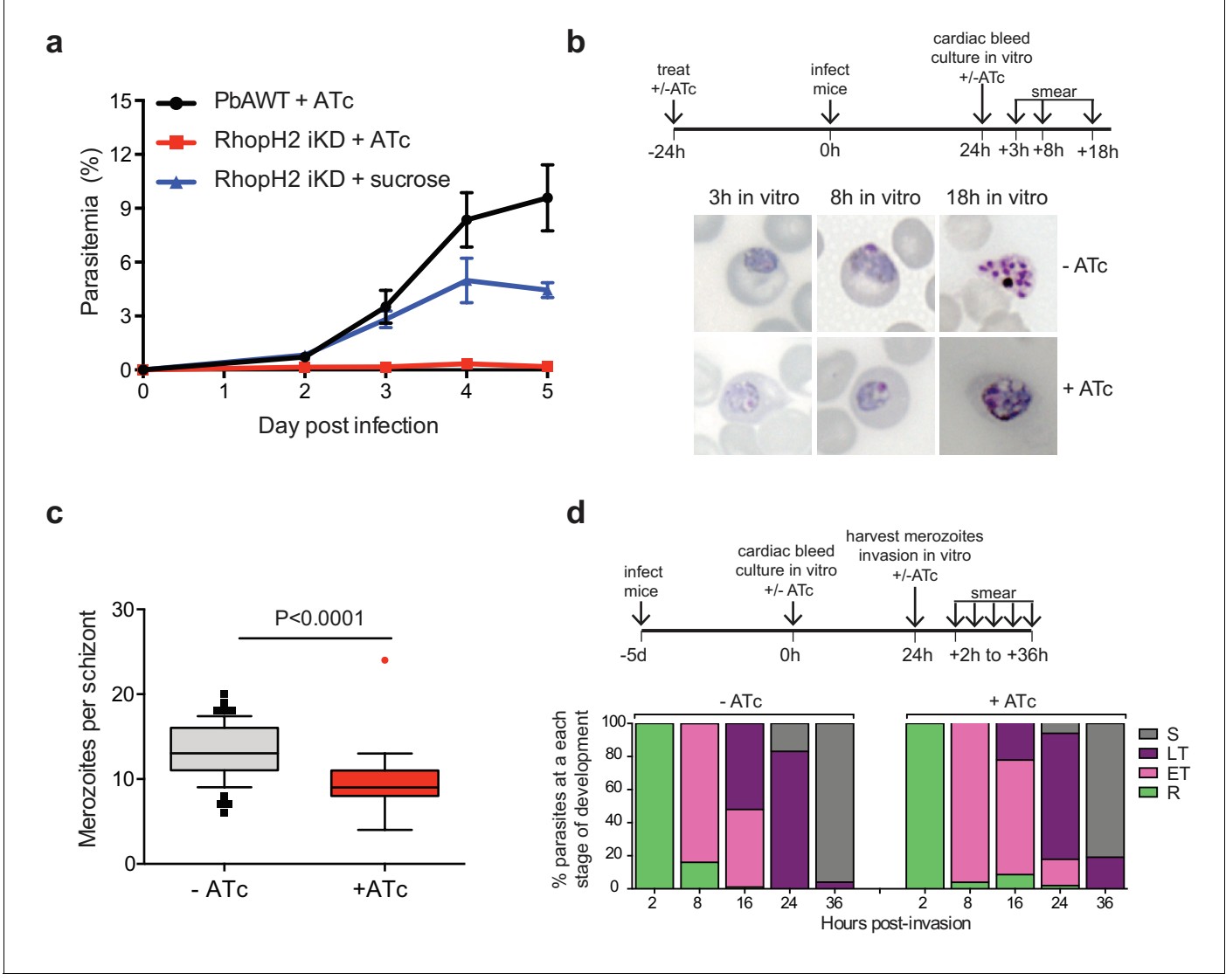

**Figure 6.** Depletion of RhopH2 in *P. berghei* leads to altered growth phenotypes in vivo and in vitro. (**a**) Representative growth curve (n = 2) of *P. berghei* iRhopH2 and wildtype (WT) parasites. Groups of 5 mice were pre-treated for 24 hr with either 0.2 mg/ml ATc or sucrose (vehicle control), then infected with the PbiRhopH2 iKD line or WT PbANKA. Parasitaemia was calculated at the indicated timepoints. Error bars represent standard error of the mean. An unpaired t-test revealed growth of RhopH iKD +ATc was significantly impaired at all time points (p<0.0001) and that of RhopH2 iKD + sucrose was slower that PbAWT +ATc by day five post infection (p=0.026) (**b**) Representative Giemsa-stained smears showing effect of RhopH2 knockdown with ATc on parasite growth and schizont formation. Schematic shows experimental outline. (**c**) Depletion of RhopH2 protein levels also impacts on the number of merozoites formed per schizont (n = 59 and 55 schizonts examined for parasites grown in the absence and presence of ATc, respectively, and taken from three individual experiments). The central bar in the box plot denotes the median whilst the whiskers delineate the 10th and 90th percentiles. p<0.0001 by unpaired t-test. (**d**) Representative invasion assay (n = 2) performed with merozoites from mechanically ruptured schizonts cultured in vitro ± ATc showing percentage of parasites from n = 50–100 that were at ring (R), early trophozoite (ET), late trophozoite (LT) or schizont (S) stage of development.

The following figure supplement is available for figure 6:

**Figure supplement 1.** Characterization of inducible *P. berghei* RhopH2 parasites.

RhopH2 could invade erythrocytes but exhibited a delay in the transition from the early to more mature trophozoite forms (*Figure 6d*), consistent with our findings in *P. falciparum*.

## Knockdown of RhopH2 does not affect protein export

RhopH2 is localized on the host cytosolic side of the PVM immediately after invasion and was found to affinity purify some components of the PTEX and a variety of exported proteins. This raised the question of whether the RhopH complex helps traffic proteins that exit PTEX through the erythrocyte cytoplasm, particularly as protein export is required to support parasite growth (*Elsworth et al., 2014*; *Dietz et al., 2014*; *Beck et al., 2014*). However, no defect in the export of PfEMP1, or trafficking of either RESA to the erythrocyte membrane or SBP1 to the Maurer's clefts was evident after knocking down RhopH2 expression with GlcN (*Figure 7a*). In contrast, the localization of RhopH3 and to a lesser extent RhopH1/clag3 was affected when RhopH2 expression was knocked down (*Figure 7—figure supplement 1*). Moreover, RhopH2-HAglmS parasites supplemented with a reduced concentration of 0.5 mM GlcN that still gave efficient RhopH2 knockdown (*Figure 7b*) and which were harvested at mid-trophozite stage before parasites growth was impaired (*Figure 7c*), could similarly export a nanoluciferase reporter (Hyp1-NLuc) as RhopH2-HAglmS (-GlcN) or 3D7 parasites (±) GlcN (*Figure 7d*).

## Knockdown of RhopH2 causes *P. falciparum*-infected erythrocytes to become resistant to sorbitol and alanine lysis

Since RhopH2 forms a complex with RhopH1, a protein implicated in NPP activity, and depletion of RhopH2 leads to growth defects around the time that NPPs are active in the infected erythrocyte, we next assessed whether RhopH2 contributes to NPP function. Sorbitol transport into infected erythrocytes requires NPP activity, resulting in hypotonicity-induced cell lysis (*Wagner et al., 2003*; *Nguitragool et al., 2011*). Thus RhopH2-HAglmS-parasitized erythrocytes transfected with the Hyp1-Nluc reporter were treated with sorbitol buffer containing NanoGlo. The degree of lysis and hence channel activity could be quantified by measuring the amount of NanoGlo hydrolysed by Hyp1-Nluc which is released during cell lysis (*Azevedo et al., 2014*). We established that GlcN-mediated knockdown of RhopH2 dramatically reduced the capacity of infected erythrocytes to be lysed by sorbitol, suggesting RhopH2 contributes to NPP activity (*Figure 8a*). In contrast, 3D7-parasitized erythrocytes treated with GlcN were not affected in their ability to be lysed by sorbitol indicating that depletion of RhopH2 and not treatment with GlcN was responsible for inhibition of NPP function (*Figure 8a*). As the ability of parasitized erythrocytes to be lysed only commences >24 hpi, Giemsa smears of parasites used in the sorbitol assays were examined but this revealed the parasites were all similarly aged (*Figure 8b*). When an iso-osmotic solution of alanine was used instead of sorbitol, similar results were obtained, with increasing concentrations of GlcN reducing the capacity of RhopH2-HAglmS parasitized erythrocytes to be lysed (*Figure 8c*). More lysis inhibition was also observed in 32 hr trophozoites compared to 24 hr trophozoites, consistent with the NPPs being more developed in older parasites.

## Knockdown of RhopH2 leads to reduced levels of vitamins and de novo synthesis of pyrimidines

Given that RhopH2 depletion appeared to affect NPP activity, we next examined the effect of depleting RhopH2 on the metabolism of *P. falciparum*-infected erythrocytes. This was undertaken by conducting comparative untargeted metabolomics on 3D7 and RhopH2-HAglmS parasites incubated in the presence and absence of 2.5 mM GlcN. Overall, ~1000 metabolites from diverse pathways were detected and assigned putative identities based on accurate mass, and confirmed using retention time where standards were available (*Creek et al., 2012*). A Principal Component Analysis (PCA) of all metabolite features across the four sample groups, 3D7 ±GlcN) and RhopH2-HAglmS (±GlcN) showed that replicates from groups 3D7 (±GlcN) and RhopH2-HAglmS (-GlcN) clustered together, and that these were metabolically different to the induced RhopH2 knockdown, RhopH2-HAglmS (+GlcN) (*Figure 9a*). This indicates that knockdown of RhopH2 causes a reproducible metabolic shift in the parasites. The heat map demonstrates a substantial impact of GlcN on global metabolite levels, even in wild-type 3D7 parasites (*Figure 9b*). Nevertheless, as indicated in the PCA analysis, the inclusion of the 3D7 (+GlcN) control allowed detection of several metabolites that were

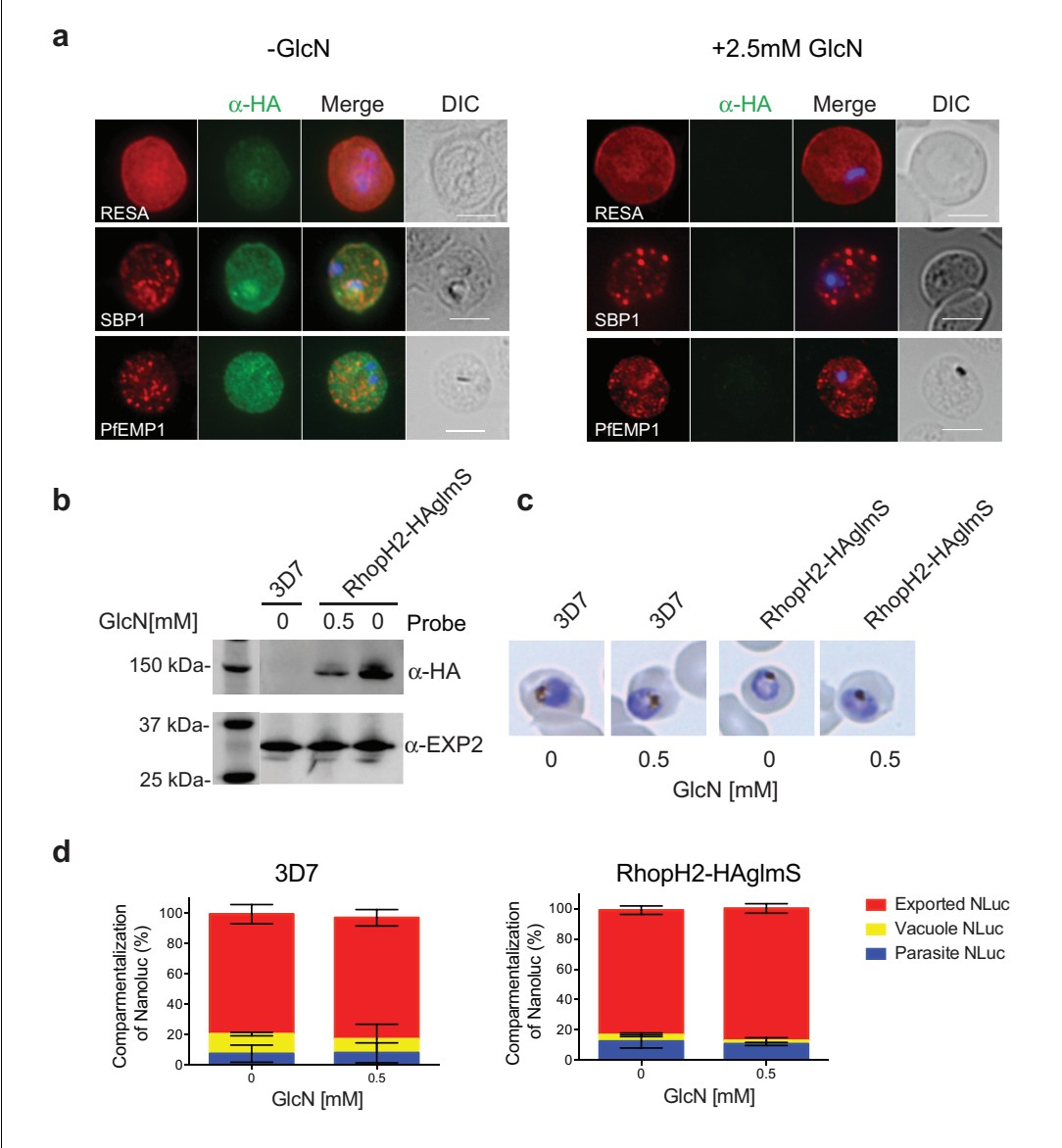

**Figure 7.** RhopH2 is not involved in the trafficking of exported proteins in the erythrocyte cytoplasm. (a) Representative IFAs of erythrocytes infected with RhopH2-HAglmS parasites grown in 0 mM or 2.5 mM GlcN using the indicated antibodies show trafficking of RESA, SBP1 and PfEMP1 is unaffected upon RhopH2 knockdown. Scale bar = 5 µm (b) Western blots of the parasites probed with an anti-HA antibody indicate that PfRhopH2 has been substantially knocked down with 0.5 mM GlcN relative to an EXP2 loading control. (c) Giemsa stained images of the trophozoites that were analysed. (d) Proportion of luciferase activity exported into the erythrocyte cytosol, secreted into the parasitophorous vacuole or present in the parasite cytoplasm of RhopH2-HAglmS and 3D7 wildtype parasites transfected with Hyp1-Nluc and grown in ± GlcN. Bars denote mean ± SD (n = 3). An unpaired t-test revealed there was no significance different in the exported NLuc fractions ± GlcN for 3D7 (p=0.8579) and RhopH2-HAglmS (p=0.1801).

The following figure supplement is available for figure 7:

**Figure supplement 1.** Localization of RhopH1/clag3 and RhopH3 in infected erythrocytes when RhopH2 expression is knocked down.

specifically perturbed in response to RhopH2 knockdown, including selected vitamins/cofactors, nucleotides, amino acids and glycolytic metabolites (*Figure 9b*). A detailed scrutiny of individual metabolites showed that while glucosamine treatment appeared to elevate metabolite levels in general, the RhopH2 knockdown resulted in decreased levels of folate and thiamin phosphates, which are essential vitamins and cofactors for cellular growth (*Figure 9c*). The other class of metabolites to

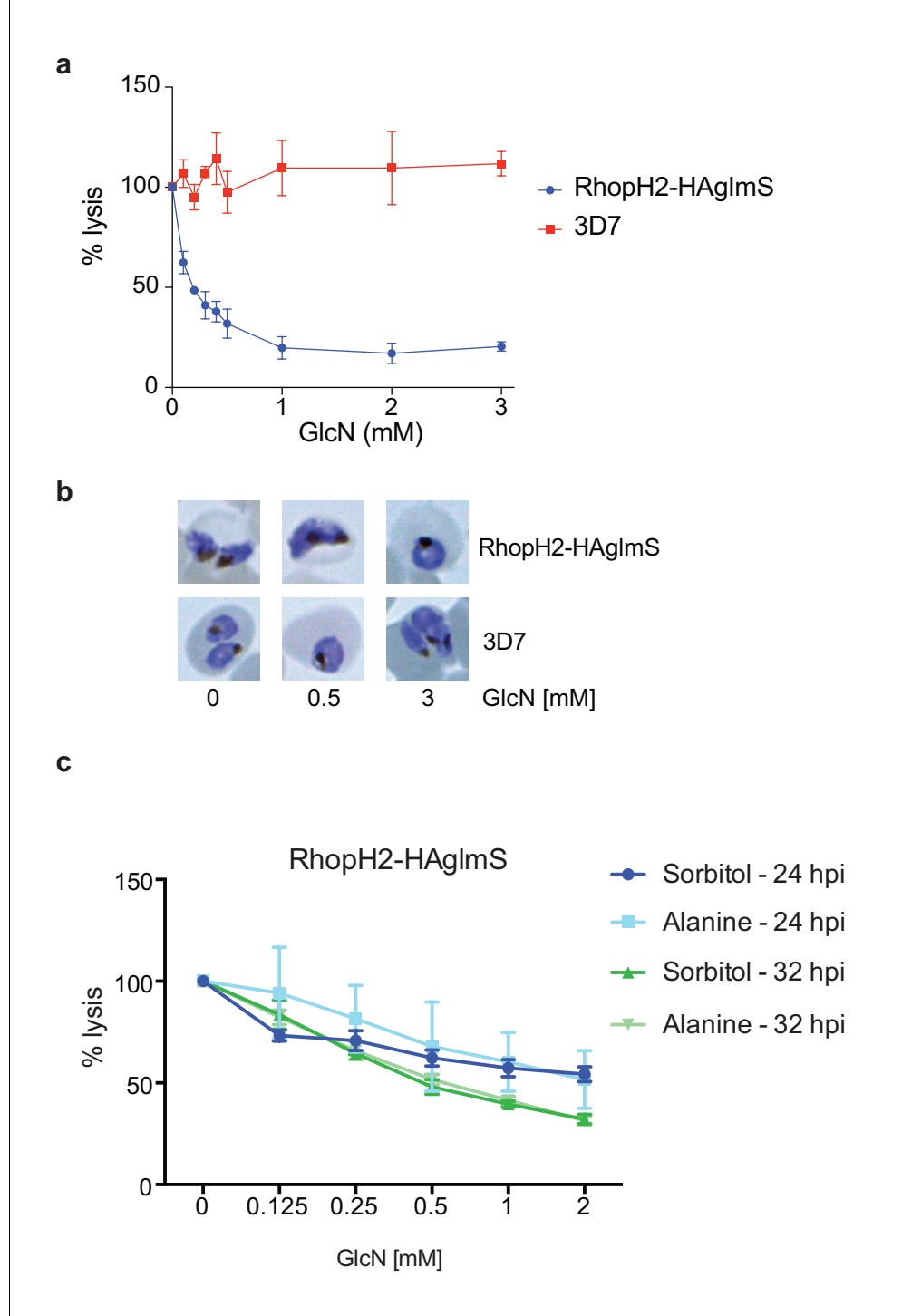

**Figure 8.** Knockdown of RhopH2 impairs sorbitol and alanine uptake. (a) GlcN-mediated knockdown of RhopH2 in PfRhopH2-HAglmS parasites expressing an exported Hyp1-Nluc reporter leads to a dramatic reduction in the capacity of infected erythrocytes to be lysed by the addition of sorbitol. In contrast erythrocytes infected with 3D7 parasites expressing Hyp1-Nluc are sensitive to sorbitol-mediated lysis. The % lysis was determined by the amount of NanoGlo substrate hydrolysed by Hyp1-Nluc, with 100% lysis defined as the Nluc activity (RLU/min) in parasites incubated in 280 mM sorbitol buffer with no GlcN. Data represents mean ± SD of three biological replicates completed in triplicate. (b) Giemsa stained images of the trophozoites analysed in the sorbitol uptake assays. (c) Analysis of sorbitol and alanine-mediated lysis of erythrocytes infected with PfRhopH2-HAglmS parasites at 24 and 32 hr post infection (hpi) at various concentrations of GlcN. The % lysis was determined by the amount of NanoGlo

*Figure 8 continued on next page*

*Figure 8 continued*

substrate hydrolysed by Hyp1-Nluc. Data represents mean ± SD of one biological experiment completed in triplicate.

significantly decrease upon RhopH2 depletion were intermediates in the de novo pyrimidine synthesis pathway, N-carbamoyl L-aspartate, dihydroorotate and orotate. These metabolites are essential nucleotide precursors in *P. falciparum*, however, levels of downstream nucleotides were not affected at this time-point (*Supplementary file 1*). Few other metabolites were significantly and specifically depleted in the RhopH2 knockdown, with the exception of the glycolytic intermediates 3-phosphoglycerate and phosphoenolpyruvate (*Supplementary file 1*). The only putatively identified metabolite to extensively accumulate (>5-fold higher than all controls) in the RhopH2 knockdown was the urea cycle intermediate argininosuccinate, however, the other urea cycle intermediates were not significantly perturbed. Interestingly, a general increase in amino acid levels was also observed in the RhopH2 knockdown (*Figure 9d*).

In order to compare these metabolic perturbations to the effect of pharmacological NPP inhibition, erythrocytes infected with 3D7 were treated with furosemide and metabolite levels compared to untreated controls. Consistent with the RhopH2 knockdown, levels of folate and phosphoenolpyruvate were significantly lower in furosemide-treated parasites, and threonine, histidine, asparagine, serine and argininosuccinate levels all increased (*Figure 9d*). Interestingly, the general depletion of de novo pyrimidine synthesis intermediates was not observed with furosemide, with N-carbamoyl L-aspartate levels found to be significantly higher following furosemide treatment.

## Discussion

In this study we have characterized RhopH2 expression and localization and investigated the consequences of knocking down RhopH2 expression on the parasite with the aim to infer function. We show that the RhopH2 synthesized during the schizont stage is carried into erythrocytes during invasion, initially localizing to the PVM. Although weak labeling of RhopH2 was also observed on the infected erythrocyte membrane, this is most likely a result of lateral diffusion of RhopH2 secreted during invasion.

From the PVM, RhopH2 then traffics through the erythrocyte cytoplasm until it reaches its final destination at the erythrocyte membrane. Although RhopH2 localizes to punctate structures in the cytoplasm, the lack of co-localization with SBP1 indicates RhopH2 is not a Maurer's clefts resident protein *per se*. This is in agreement with a previous report (*Vincensini et al., 2008*) but contrary to other studies that implicate RhopH2 as a Maurer's cleft protein (*Sam-Yellowe et al., 2001*; *Vincensini et al., 2005*).

Why the RhopH complex needs to be secreted from the rhoptries to gain access to the host cell is intriguing given the parasite has a mechanism for proteins to traverse the PVM via PTEX that is already operational shortly after invasion (*Riglar et al., 2011*; *Elsworth et al., 2014*). We originally hypothesized that the RhopH complex localizes to the host cytosolic side of the PVM immediately after invasion to act in concert with PTEX to translocate the proteins that are exported very early across the PVM. Hence, secretion of the RhopH complex via the rhoptries may provide a more timely mechanism for the complex to localize to the cytosolic face of the PVM rather than traversing through PTEX and would also allow the complex to remain intact. However, knocking down RhopH2 did not affect the export of RESA, SBP1, PfEMP1 or the Hyp1-Nluc reporter protein, while the export of these native proteins is blocked by knocking down PTEX function (*Elsworth et al., 2014*; *Beck et al., 2014*). Our results also indicate that the RhopH complex does not operate as a trafficking complex in a manner independent of PTEX to escort exported proteins throughout the erythrocyte cytoplasm (*Ling et al., 2004*).

Upon arriving at the erythrocyte membrane, RhopH2 associates with the host cytoskeleton through direct or indirect interactions with spectrin and/or ankyrin, band 3, protein 4.1 or protein 4.2, which are involved in tethering spectrin to the erythrocyte membrane. Interestingly, other proteins known to interact with the erythrocyte cytoskeleton, including RESA, MESA, and PHIST proteins also affinity purified with RhopH2. The PHIST proteins LyMP (Pf3D7_0532400) and

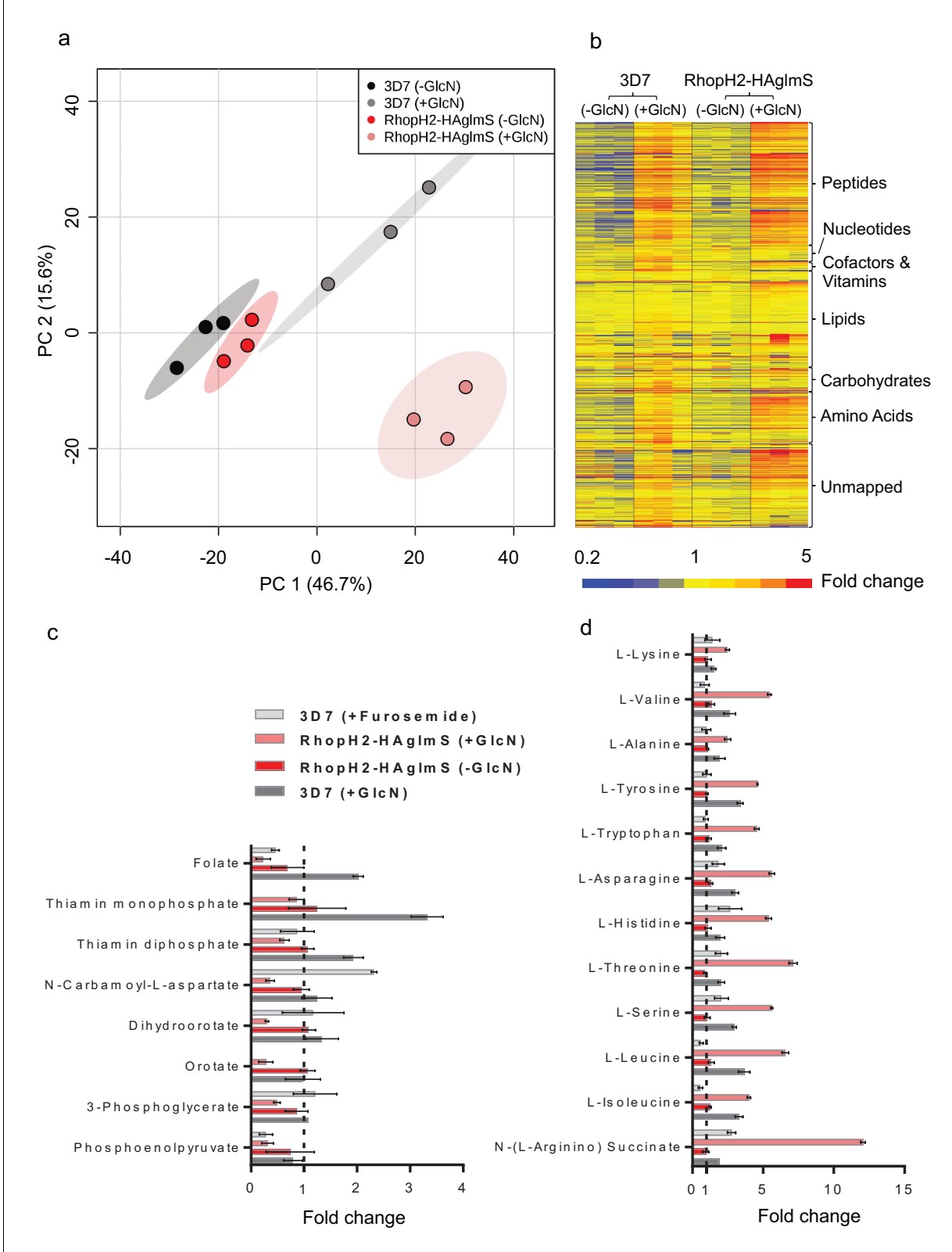

**Figure 9.** Metabolomics analysis of 3D7 and RhopH2-HAglmS parasites +/- GlcN treatment. (a) Principal Component Analysis scores plot of the first two principal components based on all metabolite features across the four sample groups. (b) Heat map of relative abundance of all the putative metabolites detected in this study grouped according to metabolite classes. (c) Fold change of metabolites showing a decrease in abundance, involved in vitamin and co-factor metabolism, de novo pyrimidine synthesis and glycolysis in the RhopH2-HAglmS (+GlcN) and 3D7 (+Furosemide) parasites

*Figure 9 continued on next page*

*Figure 9 continued*

compared to 3D7 (untreated) represented by the dotted vertical line. Error bars indicate relative standard deviation from n = 3 independent biological replicates. Thiamine monophosphate and orotate were not detected in the furosemide treatment experiment. (d) Fold change of metabolites (amino acids and a urea cycle intermediate) showing an increase in abundance in the RhopH2-HAglmS (+GlcN) and 3D7 (+Furosemide) parasites compared to 3D7 (untreated) represented by the dotted vertical line. Error bars indicate relative standard deviation from n = 3 independent biological replicates.

PF3D7_0936800 have recently been shown to interact with the acidic terminal sequence of PfEMP1 to connect this major virulence factor to the cytoskeleton (*Oberli et al., 2014*; *Proellocks et al., 2014*; *Oberli et al., 2016*). Another protein that is partially translocated to the erythrocyte membrane is SEMP1 and proteins shown to interact with SEMP1, including HSP70-x, GBP130 and the PHIST proteins PF3D7_0532300 and PF3D7_0702500 (*Dietz et al., 2014*) also interact with RhopH2.

Knocking down the expression of RhopH2 had multiple consequences for the parasite in the cycle after the addition of GlcN or ATc. The growth of the parasites was affected, particularly when they transitioned to the trophozoite stage, which resulted in a longer cell cycle for those that remained viable and hence delayed egress. By the end of the second cycle, schizonts formed fewer merozoites and those that did form appeared clumped and tethered to one another as though cytokinesis had not kept pace with the egress developmental program that triggers breakdown of the PVM and erythrocyte membranes. The failure to maintain a synchronized developmental program could be indicative of stress, perhaps due to a decrease in access to nutrients. It is, therefore, feasible that knocking down RhopH2 affected hypotonic-induced cell lysis by sorbitol, which requires NPP activity.

The reduction in the levels of folate in the RhopH2 knockdown was also striking and implicates RhopH2 in nutrient uptake. Although *P. falciparum* is capable of folate synthesis, it also needs to import exogenous folate (*Krungkrai et al., 1989*; *Wang et al., 1999*), and encodes for two folate transporters FT1 (PF3D7_0828600) and FT2 (PF3D7_1116500), which localize to the plasma membrane. RhopH2 and FT1 have very similar expression profiles (*Aurrecoechea et al., 2009*), with maximal expression observed in the late trophozoite and schizont stage. It is conceivable that RhopH2 and FT1 act in concert to facilitate the transport of folate across the erythrocyte membrane, and parasite plasma membrane, respectively. A similar decrease in folate levels observed in furosemide-treated parasites provides support for implication of RhopH2 in NPP mediated nutrient uptake. A decrease in levels of other essential vitamins and cofactors also supports the role of RhopH2 in nutrient uptake. We also saw a decrease in intermediates of pyrimidine biosynthesis in the RhopH2 knockdown, although the levels of pyrimidine precursors (aspartate and glutamine) and end products (pyrimidine nucleotides) were unchanged, indicating that pre-existing nucleotide pools are not exhausted at the stage analyzed (mid-trophozoites in the second cycle post GlcN treatment). In contrast, we saw an increase in the levels of N-carbamoyl aspartate in furosemide-treated parasites, and it was not expected that levels of this intermediate of de novo pyrimidine synthesis would directly depend on NPPs. Interestingly, other NPP inhibitors have been shown to also inhibit dihydroorotate dehydrogenase (DHODH) (*Dickerman et al., 2016*), an essential enzyme in this pathway that would modulate N-carbamoyl aspartate concentration, and it is likely that the furosemide treatment (at a concentration of 500 µM for ~24 hr) also has secondary effects on metabolism that differ from the RhopH2 knockdown. Nevertheless, the specific metabolic profile observed in both the RhopH2 knockdown and furosemide-treated parasites (i.e decreased folate and phosphoenolpyruvate, increased threonine, histidine, asparagine, serine and argininosuccinate) supports a common impact on parasite biochemistry. This metabolic profile was not observed following treatment with 100 other antimalarial compounds using the same metabolomics methodology (*Creek et al., 2016*), suggesting this profile is specific for NPP inhibition and it is consistent with the increased threonine and histidine levels reported for other NPP inhibitors (where folate, phosphoenolpyruvate, serine and argininosuccinate were not assayed) (*Dickerman et al., 2016*).

The combined reductions in folate uptake (an essential cofactor for thymine nucleotide synthesis) and de novo pyrimidine synthesis in RhopH2 depleted parasites is likely to lead to decreased availability of pyrimidine nucleotides once pre-existing pools are depleted, which may explain the reduced number of merozoites and delayed growth phenotype observed in the RhopH2 knockdown. This phenotype has been reported earlier in metabolically compromised *P. berghei* parasites with a

disrupted pyrimidine synthesis pathway (*Srivastava et al., 2015*). The mechanism responsible for the observed down-regulation of pyrimidine synthesis and glycolysis is not clear, but may be secondary to a starvation response or compromised viability. Metabolites in these two pathways are particularly susceptible to depletion in parasites exposed to a range of antimalarial compounds (*Creek et al., 2016*). The observed increase in amino acids in the RhopH2 knockdown could be due to a reduced efflux of excess amino acids produced by haemoglobin digestion (*Krugliak et al., 2002*), which may otherwise render the infected cells susceptible to osmotic challenge or, alternatively, increased protein degradation to survive nutrient starvation, in a manner analogous to that observed following isoleucine starvation (*Babbitt et al., 2012*). Whilst some consistencies with isoleucine-starved parasites were observed, it is important to note that the metabolomic impact of RhopH2 knockdown does not match directly with the metabolomic profile reported for isoleucine starvation (*Babbitt et al., 2012*), and that isoleucine levels in RhopH2 knockdown parasites were not significantly different from wild-type 3D7 (+GlcN) parasites.

It should also be noted that depletion of RhopH2 in *P. berghei* had a drastic consequence on parasite growth in mice, but when the parasites were cultured ex vivo the growth delays and aberrant parasite morphology were always less striking. This may be because the nutrients supplied to the parasites in culture are likely to be in greater abundance than what is available to the parasites in vivo (*Pillai et al., 2012*).

That RhopH2 can be extracted from membranes using carbonate provides little support for RhopH2 being an integral membrane protein and a channel in the erythrocyte membrane through which solutes are transported. Whilst RhopH1/Clag3 has a number of properties consistent with channel formation (*Nguitragool et al., 2011*, *2014*), it would be interesting to determine whether any of the parasite proteins found to interact with RhopH2 by proteomics could potentially serve as a channel component and if remodeling of the erythrocyte cytoskeleton also contributes to solute transport. By Blue-Native PAGE, RhopH2 was observed in an ~670 kDa complex—this complex is similar in size to that identified by Zainabadi (*Zainabadi, 2016*) which also comprises RhopH1/Clag3 (but not RhopH3). Interestingly, we also observed RhopH2 in an ~410 kDa complex, which is different to the ~480 kDa complex that comprises only RhopH1/Clag3 (*Zainabadi, 2016*). Whether this complex comprises of other proteins that could be affinity purified with RhopH2 remains to be ascertained.

In summary, our work reveals that the RhopH complex interacts with components of the erythrocyte cytoskeleton as well as numerous exported proteins that are involved in host cell remodeling and a schematic illustrating how the RhopH complex may traffic to the erythrocyte surface is provided in *Figure 10*. We provide the first direct genetic evidence that depletion of a member of the RhopH complex leads to altered NPP function, and depletion of essential vitamins and cofactors. The alteration to parasite growth and metabolism, as well as the effect on parasite replication and delayed egress, are in keeping with the NPPs being an important erythrocyte modification induced by the parasite. Delineating the molecular makeup of the NPPs is critical for identifying the best strategies for targeting this pathway with anti-malarial drugs as well as understanding the mechanisms by which malaria parasites can potentially alter NPPs to develop resistance to particular chemotherapeutic agents.

## Note added in proof

A manuscript utilizing the same strategy to deplete RhopH2 in *Plasmodium falciparum* was also published by Ito and colleagues (*Ito et al., 2017*). These authors derive very similar conclusions about the role of RhopH2 in nutrient uptake.

## Materials and methods

### Ethics approval

Experiments involving the use of animals were performed in accordance with the recommendations of the Australian Government and the National Health and Medical Research Council Australian code of practice for the care and use of animals for scientific purposes. The protocols were approved by the Deakin University Animal Welfare Committee (approval number G37/2013).

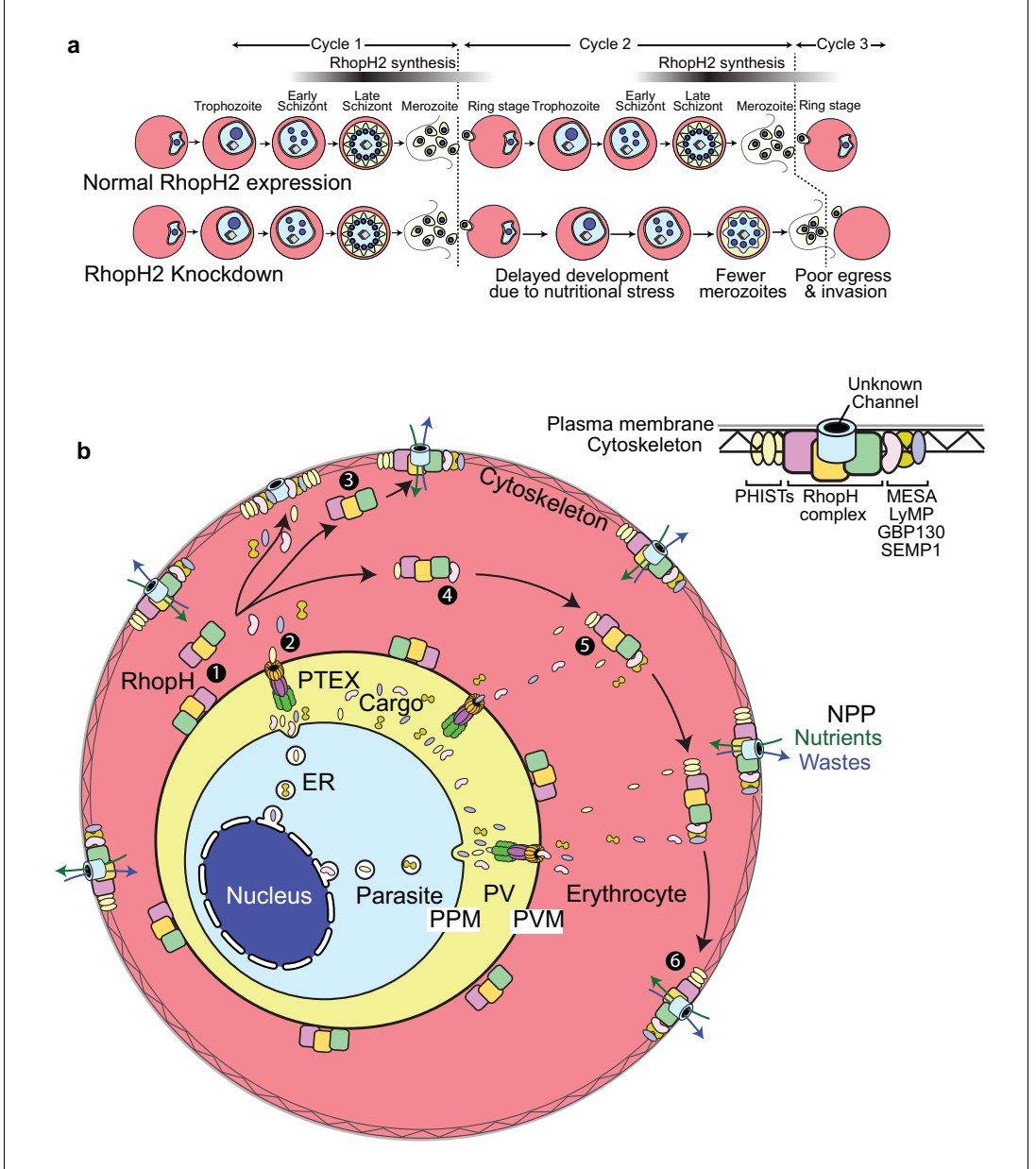

**Figure 10.** Scheme illustrating how RhopH2 knockdown effects blood stage development. (**a**) Knockdown of RhopH2 expression in cycle one appears to impair uptake of plasma nutrients in cycle two which delays development and replication in cycles 2 and 3. (**b**) (1) The RhopH complex is probably introduced onto the surface of the parasitophorous vacuole membrane (PVM) during merozoite invasion. (2) Shortly after invasion the PTEX complex begins exporting parasite-synthesised proteins secreted into the parasitophorous vacuole (PV), out into the erythrocyte cytoplasm. Some of the exported proteins such as PHISTs, MESA, LyMP, GBP130 and SEMP1 travel and bind to the erythrocyte cytoskeleton. The RhopH complex either (3) binds to these exported proteins at the erythrocyte surface or (4, 5) assembles with these proteins en route to the surface. (6) Once at the surface, the RhopH/exported protein complex forms NPPs either by forming their own membrane-spanning pore or by 'opening up' an erythrocyte pore. The NPPs function to permit the entry of nutrients and to dispose of parasite waste products.

## Plasmid constructs

To create a transgenic *P. falciparum* line in which RhopH2 expression could be knocked down, the transfection construct pRhopH2-HAglmS was created. This construct contains 1035 bp of sequence immediately upstream of the stop codon of RhopH2 (Pf3D7_0929400) that had been PCR amplified from *P. falciparum* 3D7 genomic DNA (gDNA) with the primers DO227 and DO228 (see

*Supplementary file 2* for oligonucleotide sequences) and cloned into the *Bgl*II and *Pst*I sites of pPfTEX88-HAglmS (*Chisholm et al., 2016*). To engineer the PbRhopH2 inducible knockdown (iKD) line, the first 1477 bp of the PbRhopH2 coding sequence (PbANKA_0830200) that had been PCR amplified with the primers DO291F and DO67R was cloned into the *Pst*I and *Nhe*I sites of the modified pPRF-TRAD4-Tet07-HAPRF-hDHFR (*Pino et al., 2012*) described in *Elsworth et al. (2014)*. Also cloned into the *Nhe*I and *Bss*HII sites of this vector were 1279 bp of the *rhoph2* 5' UTR sequence immediately upstream of the RhopH2 start codon, which had been PCR amplified using the primers DO62F and DO63R. Before transfection into *P. berghei* ANKA parasites, pTRAD4-iRhopH2ss was linearized with *Nhe*I.

## Parasites and transfection

Blood-stage *P. falciparum* strain 3D7 was cultured continuously (*Trager and Jensen, 1976*) and transfected as previously described (*Fidock and Wellems, 1997*). Transgenic parasites were selected with 2.5 nM WR99210 (Jacobus) or 5 µg/mL blasticidin S (Sigma-Aldrich, Australia). *P. berghei* transgenic parasites were generated using the reference clone 15cy1 from the *P. berghei* ANKA strain. Transfection of parasites and selection of the transgenic parasites intravenously injected into 6- to 8-week-old female BALB/c mice was performed as previously described (*Janse et al., 2006*).

## Analysis of RhopH2 expression levels in *P. falciparum*

Erythrocytes infected with PfRhopH2-HAglmS parasites were treated at ring stage with 2.5 mM glucosamine (GlcN) or 0 mM GlcN as a control (day 1). Parasites were harvested at schizont stage, or in the following cycle at mid ring stage or schizont stage and treated with 0.05% saponin to remove haemoglobin. Western blots of parasite proteins fractionated on 8% Bis-Tris gels (Life Technologies, Carlsbad, CA, USA) were blocked in 5% skim milk in PBS and then incubated with mouse anti-HA (1:1000; Roche, Indianapolis, IN, USA) for detection of RhopH2 and rabbit anti-EXP2 (1:1000) as a loading control. After washing, the membranes were probed with horseradish peroxidase-conjugated secondary antibodies (1:5000; Thermo Scientific, Waltham, MA, USA) and detection was performed using the Clarity ECL Western blotting substrate (Biorad, Hercules, CA, USA). The membrane was imaged using a Fujifilm LAS-4000 Luminescent Image Analyzer and ImageJ software (NIH, version 1.46r) was used to measure intensity of bands.

## Analysis of RhopH2 expression levels in *P. berghei*

Mice infected with erythrocytes infected with the PbRhopH2 iKD line were administered drinking water containing 0.2 mg/ml ATc (Sigma) made in 5% sucrose or 5% sucrose only as vehicle control when the parasitemia reached ~1%. After 24 hr when the parasites were predominantly at ring stage, mouse blood was harvested by cardiac bleed and cultured in vitro until parasites reached schizont stage (~16 hr) in RPMI 1640 medium containing L-glutamine (Life Technologies) supplemented with 25 mM HEPES, 0.2% bicarbonate, 20% fetal bovine serum and 1 µg/ml ATc (or vehicle as a control) at 36.5°C. Experiments were performed on two independent occasions. The infected erythrocytes were lysed with 0.05% saponin prior to RNA extraction. To detect transcripts in *P. berghei* parasites by qRT-PCR, RNA was extracted from blood stage parasites using TRIsure reagent (Bioline, UK). cDNA was then made using the iScript reverse transcription supermix (Biorad) according to the manufacturer's instructions. cDNA (or gDNA as a control) was used in PCR reactions using oligonucleotides to *rhopH2* (O614F/O615R and O605F/O616R) or *gapdh* (O567F/O568R). The expression levels of *rhopH2* were normalized against the *gapdh* house-keeping gene, with gene expression values calculated based on the $2^{\Delta\Delta Ct}$ method.

## Analysis of knockdown of RhopH2 expression on *P. falciparum* growth

For analysis of *P. falciparum* growth, erythrocytes infected with PfRhopH2-HAglmS parasites at ring stage were sorbitol synchronized twice and the following cycle (Cycle 1), 2.5 mM GlcN was added to ring stage parasites, with 0 mM GlcN serving as the negative control. Parasitemias in Giemsa-stained smears were determined by counting a minimum of 1000 erythrocytes and comparative growth analysis was performed using a student's t-test. Parasite growth of triplicate samples was also assessed using a modified Malstat assay protocol (*Makler and Hinrichs, 1993*). For this, GlcN (Sigma) was added to blood cultures of synchronized PfRhopH2-HAglmS ring stage parasites (~5% parasitemia

and 2% hematocrit) in cycle 1. In cycle 1 and cycle 2, when parasites were at trophozoite stage, three aliquots were removed for subsequent proliferation assays and the cultures then diluted 1/5 to 1/10 and seeded into new plates with fresh erythrocytes and GlcN. A final three aliquots were removed in cycle three when parasites were at trophozoite stage. To quantitate parasite biomass, 30 µL of culture was mixed with 75 µL Malstat reagent (0.1 M Tris pH 8.5, 0.2 g/mL lactic acid, 0.2% v/v Triton X-100 and 1 mg/mL acetylpyridine adenine dinucleotide (Sigma), 0.01 mg/ml phenozine ethosulfate (Sigma) and 0.2 mg/mL nitro blue tetrazolium (Sigma). Once the no drug control wells had developed a purple color the absorbance was measured at 650 nm in a spectrophotometer. The cumulative absorbance values were calculated by subtracting the absorbance of uninfected erythrocytes from infected erythrocytes and multiplying by the combined dilution factor. The pLDH activities of 3D7 and RhopH2-HAglmS cultured in the absence of GlcN at cycle three were normalized to 100%, and activity of all lines (± GlcN) at each day was measured relative to this. Data was analysed using a student's t-test.

## Analysis of knockdown of RhopH2 expression on *P. berghei* growth

Female Balb/c mice at 6 weeks of age were randomized into groups of five mice per experiment and then given drinking water containing either 0.2 mg/mL ATc (Sigma) made in 5% (w/v) sucrose or 5% sucrose only as a vehicle control. After 24 hr pre-treatment, mice were infected intraperitoneally (i.p) with $1 \times 10^6$ PbRhopH2 iKD parasitized erythrocytes. From 3 days post infection, parasitemias were monitored daily by Giemsa-stained tail blood smears, with mice humanely culled once the parasitemias reached >20%. Parasitemias in Giemsa-stained smears were determined by counting a minimum of 1000 erythrocytes. Comparative growth experiments were analyzed using a students *t*-test, with p<0.05 considered significant. To establish synchronous *P. berghei* infections, blood was harvested from donor mice infected with PbRhopH2 iKD when the parasitemia was ~3%. The blood was then cultured overnight in vitro in RPMI/20% FCS in the presence or absence of 1 µg/mL ATc until parasites reached schizont stage. The schizonts were purified on Nycodenz ( ELITech Group, Australia) and isolated merozoites were incubated with uninfected erythrocytes in vitro as previously described and invasion allowed to proceed for 30 min (*Matthews et al., 2013*). Following merozoite invasion, parasites were maintained in culture for a further 36 hr, with smears made at intervals and stained with Giemsa to monitor parasite growth.

## Solubility assays

Erythrocytes infected with PfRhopH2-HAglmS at either ring or schizont stage were lysed with 0.05% (w/v) saponin in PBS. For sequential solubility assays, the pelleted parasite material was resuspended in a hypotonic lysis buffer (1 mM HEPES, pH 7.4) and after a 30 min incubation on ice, the material was centrifuged at 100,000 *g* for 30 min at 4°C. The supernatant, which contains soluble proteins, was removed and kept for analysis. The pellet was then resuspended in 0.1 M $Na_2CO_3$ (pH 11.5) to extract proteins peripherally-associated with membranes. After another 30 min incubation on ice and centrifugation step, the pellet was resuspended in 1% (w/v) Triton X-100 in PBS and incubated at room temperature for 30 min to extract integral membrane proteins and re-centrifuged. The starting material, soluble fractions and the Triton X-100 insoluble fraction were electrophoresed by SDS-PAGE and transferred to nitrocellulose membrane for Western blotting. In an alternative approach, parasitized erythrocytes that had been hypotonically lysed with 1 mM HEPES, pH7.4 to remove soluble proteins were split into five equal fractions and resuspended in either 10 mM Tris-HCl, 0.1 M $Na_2CO_3$ (pH 11.5), 2% Triton X-100, 6 M urea (extracts peripheral and soluble proteins) or 2% SDS (solubilizes membrane proteins). Samples were incubated on ice for 1 hr and then centrifuged at 100,000 *g* for 30 min at 4°C. Pellet fractions were washed in 10 mM Tris-HCl. Both the soluble and insoluble fractions were analysed by Western blotting using mouse anti-HA (1:1000), rabbit anti-EXP2 (1:1000), rabbit anti-HSP101 (1:1000) and rabbit anti-SERA5 (1:1000).

## Immunoprecipitation and mass-spectrometry

Immunoprecipitations were performed on synchronised ring stage and trophozoite *P. falciparum* RhopH2-HAglmS-infected erythrocytes harvested with 0.05% (w/v) saponin in PBS. Parasite pellets were solubilized in 1% (w/v) Triton X-100 containing Complete protease inhibitors (Roche). After a 30 min incubation on ice, the material was centrifuged at 17,000 *g* for 10 min at 4°C and

supernatants were added to 100 μl PBS-washed anti-HA-agarose beads (mAb clone HA-7) (Sigma) and mixed overnight at 4°C. The beads were washed in 0.5% Triton X-100 in PBS plus protease inhibitors. Bound proteins were eluted with 100 μL 1x non-reducing sample buffer (50 mM Tris-HCl pH 6.8, 10% glycerol, 2 mM EDTA, 2% SDS, 0.05% bromophenol blue), then reduced and electro-phoresed by SDS-PAGE. After staining the gel with Imperial Protein Stain (Thermo Scientific), protein bands were manually excised and subjected to manual in-gel reduction, alkylation, and tryptic diges-tion, and extracted peptides were analysed by LC-MS/MS using an Orbitrap Lumos mass spectrome-ter (Thermo Scientific) fitted with nanoflow reversed-phase-HPLC (Ultimate 3000 RSLC, Dionex, Australia). The nano-LC system was equipped with an Acclaim Pepmap nano-trap column and an Acclaim Pepmap RSLC analytical column. 1 μL of the peptide mix was loaded onto the enrichment (trap) column at an isocratic flow of 5 μL/min of 3% $CH_3CN$ containing 0.1% formic acid for 6 min before the enrichment column was switched in-line with the analytical column. The eluents used for the LC were 0.1% v/v formic acid (solvent B) and 100% $CH_3CN$/0.1% formic acid v/v. The gradient used was 3% B to 20% B for 95 min, 20% B to 40% B in 10 min, 40% B to 80% B in 5 min and maintained at 80% B for the final 5 min before equilibration for 10 min at 3% B prior to the next sample. The mass spectrometer was equipped with a NanoEsi nano-electrospray ion source (Thermo Fisher) for automated MS/MS. The resolution was set to 120000 at MS1 with lock mass of 445.12003 with HCD Fragmentation and MS2 scan in ion trap. The top 3 s method was used to select species for fragmentation. Singly charged species were ignored and an ion threshold triggering at 1e4 was employed. CE voltage was set to 1.9 kv.

## Blue-Native PAGE

Late trophozoite-stage (24–36 hr post invasion [hpi]) *P. falciparum*-infected erythrocytes were lysed in 0.09% saponin in 5 mM Tris pH 7.5 and washed three times in PBS to remove haemoglobin. Fol-lowing centrifugation, the parasite pellet was solubilized by sonication in 0.25% (v/v) Triton X-100 or 1% (v/v) ASB-14 (3-(tetradecanoylamidopropyl dimethylammonio) propane 1-sulfonate), the latter because it is often used for solubisation of proteins for 2D electrophoresis), then incubated with mix-ing at 4°C for 30 min. Insoluble material was pelleted (14 000 g for 30 min at 4°C). The supernatants were electrophoresed on NativePAGE Novex 3–12% Bis-Tris protein gels as per manufacturer's instructions (Invitrogen) and transferred to PVDF for Western blotting. Bound antibody probes were detected with LiCor Odyssey Fc infrared imager followed by analysis with ODYSSEY v1.2 software.

## Indirect immunofluorescence analysis (IFA)

IFA was performed on thin smears of infected erythrocytes fixed with ice cold 90% acetone/10% methanol for 2 min. Cells were blocked in 1% (w/v) BSA/PBS for 1 hr. All antibody incubations were performed in 0.5% (w/v) BSA/PBS. Primary antibodies for *P. falciparum* were used at the following concentrations: rat anti-HA (1:100, Life Technologies), mouse anti-HA (1:250, Life Technologies), chicken anti-HA (1/200, Abcam), rabbit anti-RhopH1/clag3 (1:200) (*Kaneko et al., 2005*), rabbit anti-RhopH3 (1:250), rabbit anti-RAMA-D (1:1000), rabbit anti-AMA1 (1:300), rabbit anti-RON4 (1:300), rabbit anti-SBP1 (1:200), mouse anti-RESA (1:1000) and mouse anti-MSP1-19 mAb 17B6 (20 μg/mL). After a one-hour incubation in primary antibody, cells were washed three times in PBS and incubated with the appropriate AlexaFluor 488/568-conjugated secondary antibodies (1:2000) for 1 hr. Cells were washed three times in PBS, and mounted with Prolong Gold Antifade reagent (Life Technolo-gies) containing 4′,6-diamidino-2-phenylindole (DAPI) (VectorLabs, Australia). Images were taken on an Olympus IX71 microscope and processed using ImageJ v1.46r.

## Live cell imaging

Ring stage PfRhopH2-HAglmS infected erythrocytes (cycle 1) were cultured at 4% hematocrit in the presence of 3 mM GlcN (or 0 mM GlcN as a control) until parasites reached the late schizont stage of cycle 2. The culture was then diluted to 0.16% in RPMI media and 2 mL of this was allowed to set-tle to produce a monolayer onto a 35 mm Fluorodish (World Precision Instruments, Sarasota, FL, USA). Live parasite imaging was performed at 37°C on a Zeiss AxioObserver Z1 fluorescence micro-scope equipped with humidified gas chamber (90% $N_2$, 1% $O_2$, and 5% $CO_2$). Late stage schizonts were observed until they looked ready to rupture and time-lapse videos were recorded with an Axio-Cam MRm camera at four frames per second. ImageJ and Prism (Graphpad, La Jolla, CA, USA) were

used to perform image and statistical analyses. Quantitation of invasion was performed using an unpaired student's t-test.

## Invasion assays

PfRhopH2-HAglmS and 3D7 parasites transfected with pHGBHRB (a plasmid encoding a GFP reporter under the expression of under the HSP86 5' UTR) (*Wilson et al., 2010*), were used for invasion assays. Tightly synchronized parasitized erythrocytes purified using a VarioMACS magnetic cell separator were mixed with erythrocytes (1:50 ratio) that had been stained with 10 µM amine-reactive fluorescent dye 7-hydroxy-9H-(1,3-dichloro-9,9-dimethylacridin-2-one) succinimidyl ester (Cell Trace Far Red DDAO-SE) (Invitrogen, Carlsbad, CA, USA) in RPMI-1640 for 1 hr at 37°C according to manufacturer's protocol. At designated time points, erythrocytes were harvested and stained for 20 min at room temperature in the dark with the DNA dye Hoechst 34580 (2 µM) (Invitrogen) made in RPMI-1640. Following a washing step, stained samples were examined using a BD FACS Canto II flow cytometer (BD Biosciences, Australia) with 100,000 events recorded. Experiments were carried out in triplicate. The collected data was analysed with FlowJo software (Tree Star, Ashland, Oregon). Data was analysed for statistical significance using an unpaired student's t-test.

## Nanoluciferase export assay

Wildtype *P. falciparum* 3D7 and RhopH2-HAglmS-infected erythrocytes were transfected with a Nanoluciferase (Nluc) (*Hall et al., 2012*) protein N-terminally appended with the N-terminus of the PEXEL protein Hyp-1 as described in (*Azevedo et al., 2014*) but containing the blasticidin deaminase gene instead of the hDHFR gene. The infected erythrocytes were sorbitol synchronized and when the parasites reached late trophozoite stage, the cultures were treated with either 0.5 mM GlcN or no GlcN for 48 hr. Infected erythrocytes were subsequently transferred to a 96 well plate at 1% hematocrit, 1% parasitemia and the GlcN concentration was maintained prior to measurement of Nluc signal. A series of wells containing infected erythrocytes lacking exported Nluc were used to control for background luminescence. When performing the assay, the control well lacking Hyp1-Nluc was spiked with recombinant Nluc (1 ng/µL) to control for Nluc quenching by haemoglobin. Subsequently 5 µL of resuspended culture was added to Greiner Lumitrac 96 well microplate in duplicate before adding 90 µL of either Background buffer (10 mM Tris phosphoric acid pH7.4, 127 mM NaCL, 5 mM Na$_2$EDTA, 5 mM DTT), Equinotoxin (EQT) buffer (Background buffer with EQT (5 µg/mL) prepared in house as per [*Jackson et al., 2007*]), EQT/saponin buffer (EQT buffer with 0.03% (w/v) saponin) or hypotonic buffer (10 mM Tris phosphoric acid pH 7.4, 5 mM Na$_2$EDTA, 0.2% NP40, 5 mM DTT), which allow differential fractionation of the infected erythrocyte. The cells were incubated for 10 min at RT to allow for lysis to occur. Following this, 5 µl of diluted NanoGlo (Promega, Australia, diluted 1: 500 in background buffer) was injected to each well, the plate shaken (700 rpm/30 s) and relative light units were then measured with CLARIOstar plate reader (BMG Labtech, Australia). Experiments were repeated on three independent occasions and two technical replicates were completed per biological replicate. The export of Nluc was calculated as follows: the mean ($\bar{x}$) was calculated before adjusting for the spike in control (His-Nluc) and subtracting the background (buffer one). Error was estimated with standard deviation (SD) and coefficient of variation (CV). Subsequently, percentage of Nluc was calculated for each compartment using the new mean:

$$\% \, Exported \, fraction = \frac{\bar{X}_{EQT}}{\bar{X}_{Hypo}} x100$$

Standard deviation for exported fraction was calculated as follows:

$$CV_{exported \, fraction} = \sqrt{CV_{EQT}{}^2 + CV_{Hypo}{}^2}$$

$$SD_{exported \, fraction} = CV \, x \, \% \, Exported \, fraction$$

$$\% \, Secreted \, fraction = \frac{(\bar{X}_{EQT} + \bar{X}_{SAP}) - \bar{X}_{EQT}}{\bar{X}_{Hypo}} x100$$

Standard deviation for secreted fraction was calculated as follows:

$$CV_{secreted\ fraction} = \frac{\sqrt{SD_{EQT}{}^2 + SD_{EQT+SAP}{}^2}}{\bar{X}_{EQT+SAP} - \bar{X}_{EQT}}$$

$$SD_{secreted\ fraction} = \sqrt{CV_{secreted\ fraction}{}^2 + CV_{Hypo}{}^2}\ x\ \%secreted\ fraction$$

$$\%\ Parasite\ fraction = \frac{\bar{X}_{Hypo} - (\bar{X}_{EQT} + \bar{X}_{SAP})}{\bar{X}_{Hypo}} x100$$

Standard deviation for parasite cytosol fraction was calculated as follows:

$$CV_{parasite\ fraction} = \frac{\sqrt{SD_{EQT+SAP}{}^2 + SD_{Hypo}{}^2}}{\bar{X}_{Hypo} - \bar{X}_{EQT+SAP}}$$

$$SD_{parasite\ fraction} = \sqrt{CV_{parasite\ fraction}{}^2 + CV_{Hypo}{}^2}\ x\ \%parasite\ fraction$$

Experiments were then combined depending on the weight of data reliability with error weighted mean and error weighted standard deviation.

Weight of data depending reliability.

Error weighted mean was calculated as follows:

$$\bar{X}_{weigthed} = \frac{\frac{Value_1}{Abs(SD_1)} + \frac{Value_2}{Abs(SD_2)} + \frac{Value_3}{Abs(SD_3)}}{Abs\left(\frac{1}{SD_1}\right) + Abs\left(\frac{1}{SD_2}\right) + Abs\left(\frac{1}{SD_3}\right)}$$

Error weighted standard deviation was calculated as follows:

$$SD_{weighted} = \sqrt{\frac{\frac{\left(Value_1 - \bar{X}_{Weighted}\right)^2}{Abs(SD_1)} + \frac{\left(Value_2 - \bar{X}_{Weighted}\right)^2}{Abs(SD_2)} + \frac{\left(Value_3 - \bar{X}_{Weighted}\right)^2}{Abs(SD_3)}}{\frac{1}{SD_1} + \frac{1}{SD_2} + \frac{1}{SD_3}}}$$

**Value** : Percentage in relative compartment, where numbers refer to three biological replicates.

**SD:** Standard deviation calculated for each compartment, where numbers refer to three biological replicates.

Data were analysed for statistical significance using a two-tailed, unpaired student's t test with equal variances.

## Sorbitol and alanine lysis experiments of RhopH2-HAglmS knockdown parasites

*P. falciparum* RhopH2-HAglmS and 3D7 parasites expressing Hyp1-Nluc were treated with 0–3 mM GlcN when parasites were at trophozoite (28–36 hr post invasion) stage and parasites were then grown for a further 48 hr until trophozoites stage. After washing the parasitized erythrocytes twice in PBS, 10 µL at 1% hematacrit and 1% parasitemia (or PBS as a control) was dispensed in triplicate into a Thermo Scientific 96 well U bottom microplate and loaded into a Clariostar luminometer (BMG labtech). To each well, 40 µL of sorbitol or alanine lysis buffer containing the NanoGlo substrate (280 mM sorbitol or 280 mM L-alanine, 20 mM Na-HEPES, 0.1 mg/ml BSA, pH 7.4, Nano-Glo [1:1000 dilution]) was added and the relative light units (RLU) measured every 3 min with gain set to 2500. The percent lysis was determined by non-linear regression, exponential growth equation as analysed by GraphPad Prism software. The PBS control was subtracted and the value multiplied by 100 to get a percentage lysis. A value of 100% lysis is defined as the Nluc activity in relative light units (RLU/min) of parasites in 280 mM sorbitol or alanine buffer with no GlcN. A value of 0% lysis is defined as the Nluc activity of parasites in PBS containing nanoglo substrate. The rate of lysis was derived from a kinetic assay measuring the increase in RLU per minute (*Dickerman et al., 2016*). Data was analysed for statistical significance using an unpaired student's t-test.

## Egress assay experiments of RhopH2-HAglmS knockdown parasites

Erythrocytes infected with RhopH2-HAglmS and 3D7 parasites expressing NLuc (as described above) were sorbitol synchronized and subsequently GlcN-treated at trophozoite stage (cycle 1). Heparin was added (100 µg/ml) to prevent any early invasion events and was subsequently removed when schizonts were observed (GlcN concentrations were maintained). Parasites were allowed to invade over a six-hour window (cycle 2) and were subsequently sorbitol synchronized prior to seeding into 96 well plates (100 µl/1% Hematocrit/1% parasitemia). Giemsa smears were taken at late schizont stage (end of cycle 2) and when rings were observed, the cultures were pelleted (500g/3 min) and 50 µl of supernatant containing released Nluc was removed for analysis (media fraction). Infected erythrocyte cell pellets were also collected. Fractions were collected every two hours for a total of 8 hr. Prior to analysis of total Nluc content, 50 µl of media containing 1% hemocrit was added to each media fraction and 50 µl of media was added to each pellet sample to maintain equivalent volumes. Each fraction was fully re-suspended and 10 µl was added to 90 µl lysis buffer (10 mM tris phosphoric acid, 5 mM $Ka_2EDTA$, 0.2% NP40, 5 mM DTT, Nano-Glo (1:1000 dilution)) in a Greiner Lumitrac 96 well microplate prior to shaking (700 pm/30 s). Relative light units were measured with a ClAR-IOstar multimode plate reader (BMG Labtech) and data was subsequently analysed using GraphPad PRISM software.

## Metabolomics

Tightly synchronized cultures of *P. falciparum* 3D7 or RhopH2-HAglmS ring stage parasites were exposed to either 0 mM or 2.5 mM GlcN in cycle one and they were harvested in the second cycle when they had sufficient haemazoin pigment (~24 hr post-invasion) to facilitate magnetic purification using a VarioMACS magnetic cell separator. For furosemide treatment, 500 µM of furosemide was added to the cultures shortly after invasion when parasites were in the early ring stages of cycle two and the cultures were harvested at ~24 hr post-invasion. Morphology of parasites was monitored by light microscopy to obtain developmentally similar stages of parasites under GlcN and furosemide treatment and untreated control cultures. Metabolism was quenched by rapidly cooling down the cultures to 4°C, culture medium was removed following centrifugation at 3000 *g* for five minutes) and metabolites were extracted from $4.5 \times 10^7$ cells using 150 µl of extraction buffer consisting of chloroform/methanol/water (1:3:1 v/v) (spiked with 1 µM PIPES, CHAPS and Tris as internal standards) followed by vortex mixing for 1 hr at 4°C. After mixing, cellular debris was removed by centrifugation at 4°C (>15000 *g* for 10 min) and the supernatant was kept at −80°C prior to analysis. Three biological replicates were prepared for each cell line and treatment. Samples were analysed by hydrophilic interaction liquid chromatography coupled to high resolution-mass spectrometry (LC-MS) according to a previously published method (*Stoessel et al., 2016*). All samples were analyzed as a single batch, in randomized order and pooled quality control samples were analyzed regularly throughout the batch to confirm reproducibility. Approximately 250 metabolite standards were analyzed immediately preceding the batch run to determine accurate retention times to facilitate metabolite identification. Additional retention times for metabolites lacking authentic standards were predicted computationally as previously described (*Creek et al., 2011*). Data was analysed using the IDEOM workflow (*Creek et al., 2011*, *2012*). Peak areas for significant metabolites were confirmed by manual integration with Tracefinder software (Thermo Scientific). Multivariate statistical analysis utilized principal component analysis (PCA) on log-transformed and auto-scaled metabolite peak intensity data using the web-based analytical tool, MetaboAnalyst (*Xia et al., 2015*). The IDEOM files containing all metabolomics data are uploaded on Figshare and can be accessed at https://figshare.com/s/c38c0a98fb01634677f6.

## Acknowledgements

We kindly thank Halina M Pietrzak for technical assistance analysing the parasite video data. We also thank Danny Wilson for provision of pHGBHRB, Ross Coppel and Osamu Kaneko for rhoptry antibodies and the Australian Red Cross for red blood cells and serum. This work was supported by a grant from the National Health and Medical Research Council (NHMRC) of Australia (Project 1082157).

# Additional information

## Funding

| Funder | Grant reference number | Author |
|---|---|---|
| National Health and Medical Research Council | 1082157 | Tania F de Koning-Ward Paul R Gilson |

The funders had no role in study design, data collection and interpretation, or the decision to submit the work for publication.

## Author contributions

NAC, Data curation, Formal analysis, Supervision, Investigation; SAC, Data curation, Formal analysis, Investigation; HEB, PRS, Formal analysis, Investigation; AS, Data curation, Formal analysis, Investigation, Visualization, Writing—review and editing; TKJ, GEW, SG, Data curation, Investigation; BSC, Validation, Project administration; DJC, Formal analysis, Supervision, Validation, Writing—review and editing; PRG, Conceptualization, Formal analysis, Funding acquisition, Validation, Investigation, Writing—original draft, Project administration, Writing—review and editing; TFdK-W, Conceptualization, Data curation, Formal analysis, Supervision, Funding acquisition, Validation, Investigation, Methodology, Writing—original draft, Project administration, Writing—review and editing

## Author ORCIDs

Natalie A Counihan, http://orcid.org/0000-0002-8973-3344
Tania F de Koning-Ward, http://orcid.org/0000-0001-5810-8063

## Ethics

Animal experimentation: Experiments involving the use of animals were performed in accordance with the recommendations of the Australian Government and the National Health and Medical Research Council Australian code of practice for the care and use of animals for scientific purposes. The protocols were approved by the Deakin University Animal Welfare Committee (approval number G37/2013).

# Additional files

## Supplementary files

• Supplementary file 1. Metabolomics analysis of 3D7 and RhopH2-HAglmS parasites ± GlcN treatment. Complete list of putative metabolites identified in this study are shown together with the fold change compared to untreated 3D7 parasites. The relative standard deviation from n = 3 independent biological replicates is also shown (see attached Excel spreadsheet).

• Supplementary file 2. Oligonucleotide sequences used in this study.

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
