## [Decision Letter]

Thank you for submitting your article "Plasmodium parasites deploy RhopH2 into the host erythrocyte to obtain nutrients, grow and replicate" for consideration by *eLife*. Your article has been favorably evaluated by Michael Marletta (Senior Editor) and three reviewers, one of whom, Dominique Soldati-Favre (Reviewer #1), is a member of our Board of Reviewing Editors.

The reviewers have discussed the reviews with one another and the Reviewing Editor has drafted this decision to help you prepare a revised submission.

Summary:

*Plasmodium falciparum* modifies host erythrocytes to render them permeable to nutrient uptake. The mechanisms underlying formation of these new permeability pathways (NPP) are not fully understood. In this study, Counihan et al. show that RhopH2 plays a critical role during Plasmodium growth inside erythrocytes, and provide data supporting a role of the RhopH complex in the function of NPPs.

The authors generated a *P. falciparum* transgenic line expressing a HA-tagged version of RhopH2 under control of the glmS ribozyme. They show by immunofluorescence that RhopH2-HA localizes to the parasite rhoptries, and is exported to the PV and beyond to the erythrocyte membrane after invasion. Using immune-precipitation combined with mass spectrometry, they identified parasite and host proteins interacting with RhopH2, including the other members of the RhopH complex (RhopH3 and clag). Using the glmS riboswitch, they could knockdown RhopH2 expression, resulting in a reduction of parasite growth and replication. A similar phenotype was observed in vivo with the rodent parasite *P. berghei* after conditional knockdown of PbRhopH2 using a tetracyclin-regulated system. The authors further show that in *P. falciparum* knocking-down RhopH2 does not affect protein export but leads to resistance of infected erythrocytes to sorbitol lysis, suggesting impairment of NPP function. Finally, a metabolomics analysis revealed several metabolites that were specifically perturbed in response to RhopH2 knockdown.

Essential revisions:

This study is very well performed, involves a large amount of data, including conditional genetics approaches combined with proteomics and metabolomics. The results provide novel and important insights into the function of a critical protein complex involved in *Plasmodium falciparum* growth inside erythrocytes. However, the three reviewers raised issues that will need to be addressed in order to improve the manuscript.

1) Evidence supporting a role of RhopH2 in NPP function is rather indirect. NPP function was assayed only with the sorbitol lysis assay, at only one time point (in cycle 2, 48 hours post-treatment with GlcN). Since RhopH2 alters parasite growth during cycle 2, one could argue that the NPP may not be functional yet in the RhopH2-HAglmS trophozoites that were analyzed. The authors should consider including a later time point in this assay, or use another assay to confirm NPP dysfunction. The metabolomics analysis reveals many metabolites are altered in RhopH2-HAglmS parasites treated with GlcN. How do these modifications compare with those induced by pharmacological inhibition of NPP? To consolidate the data on NPP dysfunction it will be important to investigate the import of at least one other solute beside sorbitol, and perform comparison of metabolome of mutant with metabolome of parasite exposed to NPP inhibitors (PMID 27874068).

2) The authors rely on the glmS system to conditionally deplete RhopH2. It is important to show that, in the absence of GlcN, RhopH2 protein expression is not altered by the fusion to HA and glmS, in comparison to the WT parental strain, and that the transgenic line grow normally. A western blot using anti-RhopH2 would be useful (in addition to anti-HA).

3) The authors characterized the phenotypic consequences of GlcN-induced knockdown of RhopH2, but provide no data on the protein expression and localization of RhopH2 and other members of the RhopH complex, especially RhopH1/clags, which have been linked to NPP function, in depleted parasites. They could use anti-RhopH3 antibodies, as in Figure 2.

---

## [Author Response]

*Essential revisions:*

*This study is very well performed, involves a large amount of data, including conditional genetics approaches combined with proteomics and metabolomics. The results provide novel and important insights into the function of a critical protein complex involved in Plasmodium falciparum growth inside erythrocytes. However, the three reviewers raised issues that will need to be addressed in order to improve the manuscript.*

*1) Evidence supporting a role of RhopH2 in NPP function is rather indirect. NPP function was assayed only with the sorbitol lysis assay, at only one time point (in cycle 2, 48 hours post-treatment with GlcN). Since RhopH2 alters parasite growth during cycle 2, one could argue that the NPP may not be functional yet in the RhopH2-HAglmS trophozoites that were analyzed. The authors should consider including a later time point in this assay, or use another assay to confirm NPP dysfunction. The metabolomics analysis reveals many metabolites are altered in RhopH2-HAglmS parasites treated with GlcN. How do these modifications compare with those induced by pharmacological inhibition of NPP? To consolidate the data on NPP dysfunction it will be important to investigate the import of at least one other solute beside sorbitol, and perform comparison of metabolome of mutant with metabolome of parasite exposed to NPP inhibitors (PMID 27874068).*

NPP function has now been investigated by examining the import of another solute, that being alanine. We have also undertaken the assays at two different time points (in cycle two at 24 h post-treatment with glucosamine and a much later timepoint at 32 h to ensure that the parasites are mature enough). In addition, we have used increasing concentrations of glucosamine (0-2.5 mM) and this shows a dose-response effect. We have now included the data in the subsection “Knockdown of RhopH2 causes *P. falciparum*-infected erythrocytes to become resistant to sorbitol and alanine lysis” and incorporated a new panel in Figure 8.

We have also performed a metabolome comparison between RhopH2-HAglmS parasites treated with glucosamine, wildtype (3D7) parasites exposed to the NPP inhibitor furosemide and untreated parasites lines as controls. We found that the key metabolic impact of RhopH2 knockdown that was attributed to a decrease in nutrient uptake is also seen when NPPs are inhibited by furosemide. We did not attempt to compare the effects of the NPP inhibitors MMV20439 and MMV007571 because these additionally block parasite DHODH which catalyses the fourth step in pyrimidine synthesis increasing N-carbamoyl-aspartate and dihydoorotate (PMID 27874068, Dickerman BK et al. 2016 Sci Rep. 2016 Nov 22; 6:37502). Furosemide treatment in 3D7 did increase N-carbamoyl-aspartate levels but not dihydroorotate suggesting DHODH was not being affected. We have now added the furosemide treatment data to Figure 9 and have accordingly altered the text in the Results and Discussion sections. This additional experiment also provided the opportunity to independently verify the results of our reported RhopH2 metabolomics study. The reported differences were observed consistently in the additional experiment, with the exception of three metabolites that we have now removed from Figure 9 (dihydroxyacetone phosphate, pantothenate and 4-phosphopantothenate).

*2) The authors rely on the glmS system to conditionally deplete RhopH2. It is important to show that, in the absence of GlcN, RhopH2 protein expression is not altered by the fusion to HA and glmS, in comparison to the WT parental strain, and that the transgenic line grow normally. A western blot using anti-RhopH2 would be useful (in addition to anti-HA).*

Despite repeated attempts to express RhopH2 (including published RhopH2 fusion proteins) we have been unsuccessful and thus have not been able to generate adequate antibodies. We requested the published RhopH2 antibody from Anthony Holder’s laboratory that was generated several decades ago but they have very limited supply and were unable to provide us with any.

However, we have data that directly compares the wildtype and transgenic *P. falciparum* parasite line for growth (in the absence of glucosamine) and have now included this data in the Results section (subsection “Modification of the rhoph2 locus in *P. falciparum*”) and in Figure 1—figure supplement 1. Statistical analysis reveals that the fusion of HA and glmS is not impacting on the ability of the transgenic line to grow normally.

With respect to the wildtype and *P. berghei* transgenic parasite lines, the inducible minimal inducible promoter does impact on the growth of RhopH2 iKD but the growth is not significantly slower than wildtype +ATc until day 5 (two-tailed unpaired t-test), whereas the growth of RhopH2 iKD +ATc compared to RhopH2 iKD +sucrose is significantly impaired at all time points when parasitemia was determined. Moreover, all phenotypic analysis of the lines (Figure 6) was performed within 2 days of RhopH2 knockdown.

*3) The authors characterized the phenotypic consequences of GlcN-induced knockdown of RhopH2, but provide no data on the protein expression and localization of RhopH2 and other members of the RhopH complex, especially RhopH1/clags, which have been linked to NPP function, in depleted parasites. They could use anti-RhopH3 antibodies, as in Figure 2.*

For these studies we used an antibody that recognizes the N-terminal of RhopH1/clag 3 previously described by Kaneko O et al. 2005 MBP 143:20-8 (PMID 15953647).

While we see good co-localisation of RhopH2 and RhopH3 at the infected erythrocyte membrane, we only saw weak-staining of RhopH1 at the erythrocyte membrane, which didn’t appear to co-localise with RhopH2. In many cases we also saw RhopH1 towards the periphery of the parasite and thus have included this in the 0mM GlcN panel. This is in contrast to signal obtained with the RhopH1 antibody that recognizes the C-terminal of RhopH1/clag 3 as published by Nguitragool et al., Cell, 2011. We have also analysed the localization of the RhopH proteins upon RhopH2 knockdown and included this data in supplemental Figure 3. This shows that knockdown of RhopH2 expression affects the localization of RhopH3 at the erythrocyte membrane and to a lesser extent that of RhopH1/clag3.